# CADmium: Fine-Tuning Code Language Models for Text-Driven Sequential CAD Design

**Prashant Govindarajan**[*,1,2,3]    **Davide Baldelli**[*,1,2,3]    **Jay Pathak**[4]    **Quentin Fournier**[2]
**Sarath Chandar**[1,2,3,5]

[1]Chandar Research Lab    [2]Mila – Quebec AI Institute    [3]Polytechnique Montréal    [4]Ansys    [5]Canada CIFAR AI Chair

**Reviewed on OpenReview:** `https://openreview.net/forum?id=lExqWvQht8`

## Abstract

Computer-aided design (CAD) is the digital construction of 2D and 3D objects, and is central to a wide range of engineering and manufacturing applications like automobile and aviation. Despite its importance, CAD modeling remains largely a time-intensive, manual task. Recent works have attempted to automate this process with small transformer-based models and handcrafted CAD sequence representations. However, there has been little effort to leverage the potential of large language models (LLMs) for sequential CAD design. In this work, we introduce a new large-scale dataset of more than 170k CAD models annotated with high-quality, human-like descriptions generated with our pipeline based on GPT-4.1. Using this dataset, we fine-tune powerful code-LLMs to generate CAD sequences represented in a JSON-based format from natural language descriptions, demonstrating the viability and effectiveness of this approach for text-conditioned CAD generation. Because simple metrics often fail to reflect the quality of generated objects, we introduce geometric and topological metrics based on sphericity, mean curvature, and Euler characteristic to provide richer structural insights. Our experiments and ablation studies on both synthetic and human-annotated data demonstrate that CADmium is able to automate CAD design, drastically speeding up the design of new objects. The dataset, code, and fine-tuned models are available online[1][2].

## 1 Introduction

Throughout history, humans have developed ever-more complex objects for survival, architectural, engineering, and artistic purposes. These range from early inventions such as hunting tools, wheels, and clocks, to modern systems such as space shuttles. With the emergence of computers, programs were written to process machining instructions, and soon after, to design 3D objects virtually. Computer-aided design (CAD) has since been widely adopted, significantly improving the speed and efficiency of the engineering and manufacturing process. However, the CAD pipeline still requires significant human effort and expertise.

Recognizing that CAD is a sequential process involving 2D sketches, such as lines and circles, followed by 3D operations like extrusions, recent studies have leveraged transformer architectures and publicly available CAD datasets to automatically generate design steps in an autoregressive manner (Wu et al., 2021; Khan et al., 2024). Recently, there has been a growing interest in generating 3D objects from natural language descriptions, as this approach would make CAD accessible to a broader, non-expert audience and speed up the creation process even further. Such an interest likely stems from the progress in large pre-trained foundation

---

[*]Equal Contribution
[1]`https://github.com/chandar-lab/CADmium`
[2]`https://huggingface.co/collections/chandar-lab/cadmium`

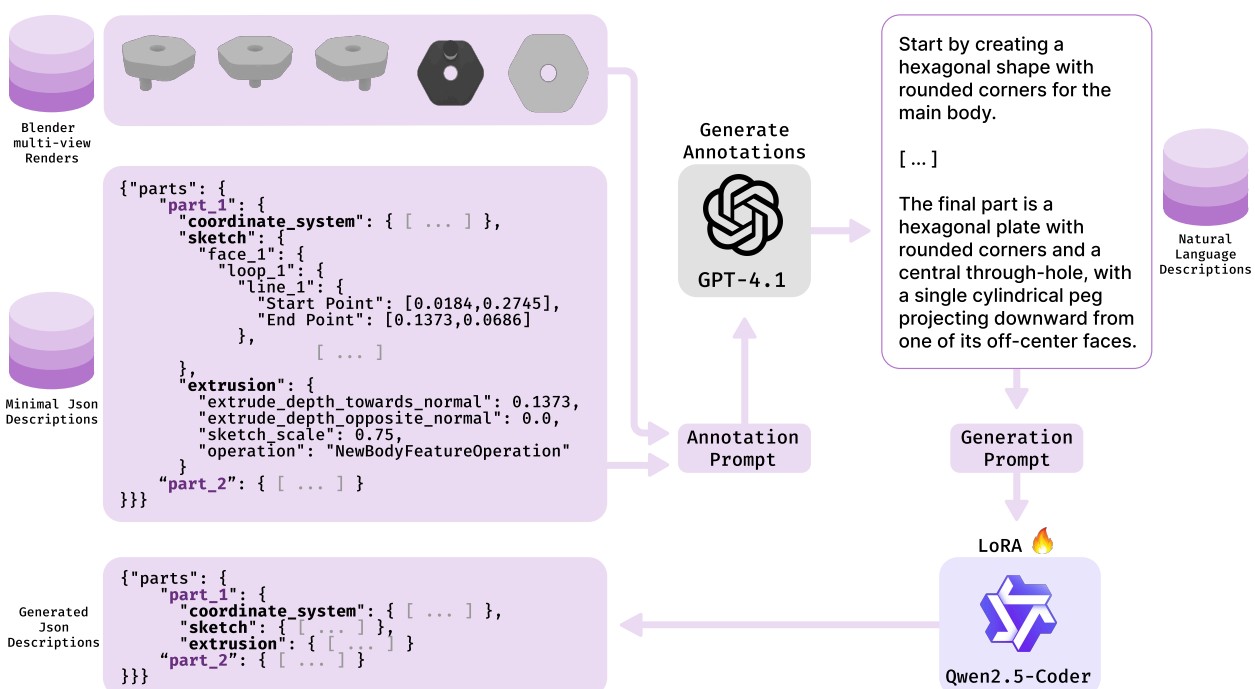

Figure 1: **The CADmium pipeline for text-to-CAD generation.** CADmium reformulates CAD generation as a purely text-to-text task. First, GPT-4.1 generates natural-sounding yet geometrically precise descriptions of 176,017 objects using their construction sequences in minimal JSON, and up to 10 multi-view images rendered with Blender. Then, the Qwen2.5-Coder LLM is fine-tuned with LoRA to translate these descriptions back into CAD sequences.

models that has resulted in tremendous reasoning and generative capabilities in natural language (Brown et al., 2020) and images (Esser et al., 2024). However, unlike these modalities, text-conditioned CAD object generation has not seen the same success as the field still faces two key challenges: (1) the lack of high-quality, human-like textual annotations for CAD samples, and (2) the absence of suitable embeddings for CAD design histories that facilitate the effective use of pre-trained language models.

Current approaches address the first challenge by automating text annotation using sophisticated state-of-the-art annotation pipelines with vision-language and mixture of experts models (Khan et al., 2024; Xu et al., 2024) and the second with vectorized representations of CAD sequences combined with separate embedding layers and prompt tokens to train language models (Khan et al., 2024; Xu et al., 2024). However, we identify critical shortcomings with both strategies. First, current machine-generated annotations often fail to strike a balance between fluency and geometric specificity, a balance essential for natural human interaction while containing sufficient geometric detail to unambiguously specify the desired 3D object. Second, prior approaches treat CAD generation as a domain-specific task requiring custom embeddings, thereby failing to fully leverage the potential of pre-trained language models and requiring significant computational resources to learn new embeddings from limited data.

To address these limitations, we introduce **CADmium**[3], a novel approach that reformulates CAD generation as a purely text-to-text task. Our contributions are as follows. First, we develop a novel annotation pipeline using GPT-4.1 (OpenAI, 2025) that processes up to 10 multi-view images of 3D objects along with their construction sequences to generate annotations that combine natural language fluency with geometric precision, improving upon prior approaches like Text2CAD (Khan et al., 2024) by leveraging a single, more capable multimodal foundation model. We apply our pipeline on 176,017 CAD sequences from Deep-CAD (Wu et al., 2021) and release a new dataset with improved expert-quality annotations. Second, we

---

[3]The name CADmium reflects the integration of "CAD" (Computer-Aided Design) and draws inspiration from the metal cadmium, known for its malleability and utility in precision applications.

fine-tune Qwen 2.5-Coder-14B (Yang et al., 2024), a state-of-the-art instruction-tuned code LLM, on our dataset to generate JSON-formatted CAD sequences from natural language prompts, leveraging the capabilities of the pre-trained model without the need for specialized embedding layers. We acknowledge that the competitive performance achieved is significantly derived from the impressive pre-trained qualities of the Qwen 2.5-Coder base model, upon which our SFT efforts provide domain-specific refinement. Recent works have shown that fine-tuning LLMs can be effective for generating code used in visual content creation (Rodriguez et al., 2024; 2025), emphasizing their versatility in visual generative applications. We evaluate our approach against state-of-the-art CAD generation models, including Text2CAD (Khan et al., 2024), using both point-cloud and mesh-based metrics established in prior work. We also validate our models on the Fusion360 Reconstruction dataset (Willis et al., 2021) using annotations generated by our pipeline and the *CADPrompt* benchmark (Alrashedy et al., 2024) that contains 200 objects annotated by human experts. Third, we introduce novel metrics based on sphericity, mean curvature, Euler characteristic, and watertightness to more extensively evaluate the structural and topological characteristics of the generated objects. We highlight the human-like fluency and conciseness of CADmium's expert-level text annotations relative to Text2CAD, supported by extensive empirical analysis. Furthermore, our fine-tuned LLM achieves competitive performance against the pre-trained Text2CAD model, with improvements across some evaluation metrics as model size increases.

## 2 Related Works

The Fusion 360 Gallery (Willis et al., 2021) was the first dataset of human-designed CAD objects to include their corresponding construction sequences. However, the Fusion 360 Gallery only contains 8,625 samples. As this was not enough for training, DeepCAD (Wu et al., 2021) introduced a significantly larger dataset of 176,017 CAD models constructed using only sketch and extrude operations parsed from the ABC dataset (Koch et al., 2019). Leveraging their large-scale dataset, DeepCAD trained a transformer-based autoencoder within a latent-GAN framework for unconditional CAD generation. SkexGen (Xu et al., 2022) and SolidGen (Jayaraman et al., 2022) later improved upon DeepCAD by introducing alternative input representations such as disentangled codebooks and boundary representations. However, these approaches were limited by their unconditional nature, offering no mechanism for users to control the generated objects.

As a result, subsequent works shifted toward conditional generation from point clouds, images, or text inputs (Xu et al., 2024; You et al., 2024; Alam & Ahmed, 2024; Uy et al., 2022; Khan et al., 2024; Dupont et al., 2024). Progress in text-conditioned CAD generation was, however, hindered by the lack of datasets with textual annotations. The first large-scale effort toward textual supervision for CAD generation was Text2CAD (Khan et al., 2024), which introduced an LLM/VLM-based annotation pipeline over DeepCAD and produced 660k beginner- to expert-level descriptions. Although foundational, its beginner-level prompts are often high-level and underspecified, while the expert-level ones tend to be overly long, redundant, or stylistically unnatural (some examples are given in Appendix A). Subsequent works expanded this direction with increasingly complex pipelines. OmniCAD (Xu et al., 2024) generated machine-produced descriptions using InternVL2-26B. CADFusion (Wang et al., 2025) also released a text-to-CAD dataset which relies on GPPT-4o and human refinement. More recently, CADLLM/TCADGen (Liao et al., 2025) proposed a semi-automated human-in-the-loop pipeline that separately processes multi-view images, point clouds, and CAD parameters to obtain detailed micro- and macro-level descriptions, producing a richer dataset of expert-style annotations. Across these efforts, existing datasets diverge into two categories: (i) expert-level, geometrically grounded descriptions (e.g., Text2CAD, CADLLM) that enable unambiguous reconstruction and form the basis of our comparisons, and (ii) higher-level, abstract, or stylistic prompts (e.g., OmniCAD, beginner prompts in Text2CAD and CADFusion) that are not directly comparable due to their limited geometric specificity. We intend to generate high-quality CAD annotations that strike a balance: they must provide sufficient geometric detail to allow precise reconstruction, while remaining concise and plausibly human-written so as to reflect real-world design prompting scenarios.

In addition to annotating the DeepCAD dataset, Text2CAD introduced a supervised generation pipeline using a frozen BERT encoder (Devlin et al., 2019), adaptive text layers, and a transformer decoder with cross-attention between text and CAD embeddings. CAD-MLLM (Xu et al., 2024) improved over this framework by incorporating LLMs and conditioning the generations on multiple modalities, namely point

clouds, images, and text. CAD-MLLM also introduced a new annotation pipeline and additional CAD samples parsed from the ABC dataset. In parallel, FlexCAD (Zhang et al., 2024) proposed a hierarchical text representation of CAD sequences to make the generation more controllable. With CADmium, we primarily aim to advance state-of-the-art performance in the text-to-CAD task, leaving multimodal integration and enhanced controllability for future work. Recent works have explored related but distinct avenues: CAD-Coder (Guan et al., 2025) focuses on image-to-CAD generation using a VLM to produce executable Python code, a different input modality and sequence representation than our text-to-text JSON-based approach. Similarly, CADFusion (Wang et al., 2025) explores distinct training strategies like combining sequential learning with visual feedback via DPO. Finally, Liao et al. (2025) proposed a dual-channel Transformer-based generator (TCADGen) alongside a refinement model (CADLLM) to convert textual design prompts into executable CAD modeling sequences.

Evaluating generative CAD models remains a challenge. Earlier works such as DeepCAD, SkexGen, and SolidGen relied on point-cloud-based metrics, including Chamfer Distance, F1 Score, Minimum Matching Distance (MMD), Coverage (COV), and Jensen-Shannon Divergence (JSD). While standard in 3D shape generation, these metrics often obscure fine structural details by uniformly sampling points throughout the object volume, failing to account for internal edges. To address these limitations, CAD-MLLM introduced mesh-based metrics such as self-intersection ratio, dangling edge length, and flux enclosure, which better capture geometric and topological correctness. Nevertheless, even these mesh-aware metrics remain limited in their ability to assess the functional quality of generated CAD models.

## 3 Methods

### 3.1 Datasets and Annotation Pipeline

We address the quality concerns of Text2CAD annotations by re-annotating all 176,017 samples from the DeepCAD library using the same multimodal inputs as Text2CAD. These inputs include the minimal JSON metadata extracted from DeepCAD using Onshape's FeatureScript (ons), which improves human readability by eliminating redundant information, and 10 multi-view images rendered with Blender's Python module (Blender Online Community, 2025) corresponding to the top, bottom, and 8 sides. We leverage the multimodal capabilities of GPT-4.1 to generate concise, natural-sounding yet geometrically precise descriptions. Additionally, we annotate 8,307 CAD objects from the Fusion 360 Reconstruction Dataset (Willis et al., 2021) not found in the ABC corpus from which DeepCAD is derived, offering a robust measure of generalization. This comprehensive re-annotation effort across both datasets involved processing a total of over 600 million input and output tokens with GPT-4.1. Importantly, only the expert-level annotations in Text2CAD provide enough geometric detail to unambiguously reconstruct objects. While it may be useful to generate plausible objects from incomplete descriptions in some cases, evaluating them is challenging as multiple outputs could satisfy a partial description. As a result, we generate exclusively expert-level prompts and restrict our comparisons to the expert-level descriptions in Text2CAD, and CADLLM annotations.

Our pipeline improves upon Text2CAD in two key ways. First, instead of using separate vision and language LLMs, we leverage GPT-4.1, a single multimodal foundation model, enabling a more accurate interpretation of input images considering the provided CAD history. Second, GPT-4.1 drastically surpasses Mistral-50B, resulting in annotations that are more human-like, concise, and often avoid technical jargon while remaining accurate, as illustrated in Figure 2.

Looking first at corpus-level statistics, CADmium's annotations are drastically more concise, most being between 100 and 200 words, compared to Text2CAD's and CADLLM's longer and often redundant annotations, often exceeding 200 words (Figure 2c). Despite being shorter, CADmium's annotations achieve a greater lexical variety, with a higher ratio of unique words per annotation (Figure 2d). The vocabulary growth curve also shows that CADmium uses a much broader range of words across annotations, reaching over 18,000 unique words or twice as many as Text2CAD, and comparably to CADLLM (Figure 2a). This richer vocabulary creates more diverse descriptions and reduces the risk of overfitting to a limited set of words. Additionally, CADmium's annotations represent numbers in a more natural, human-like way compared to

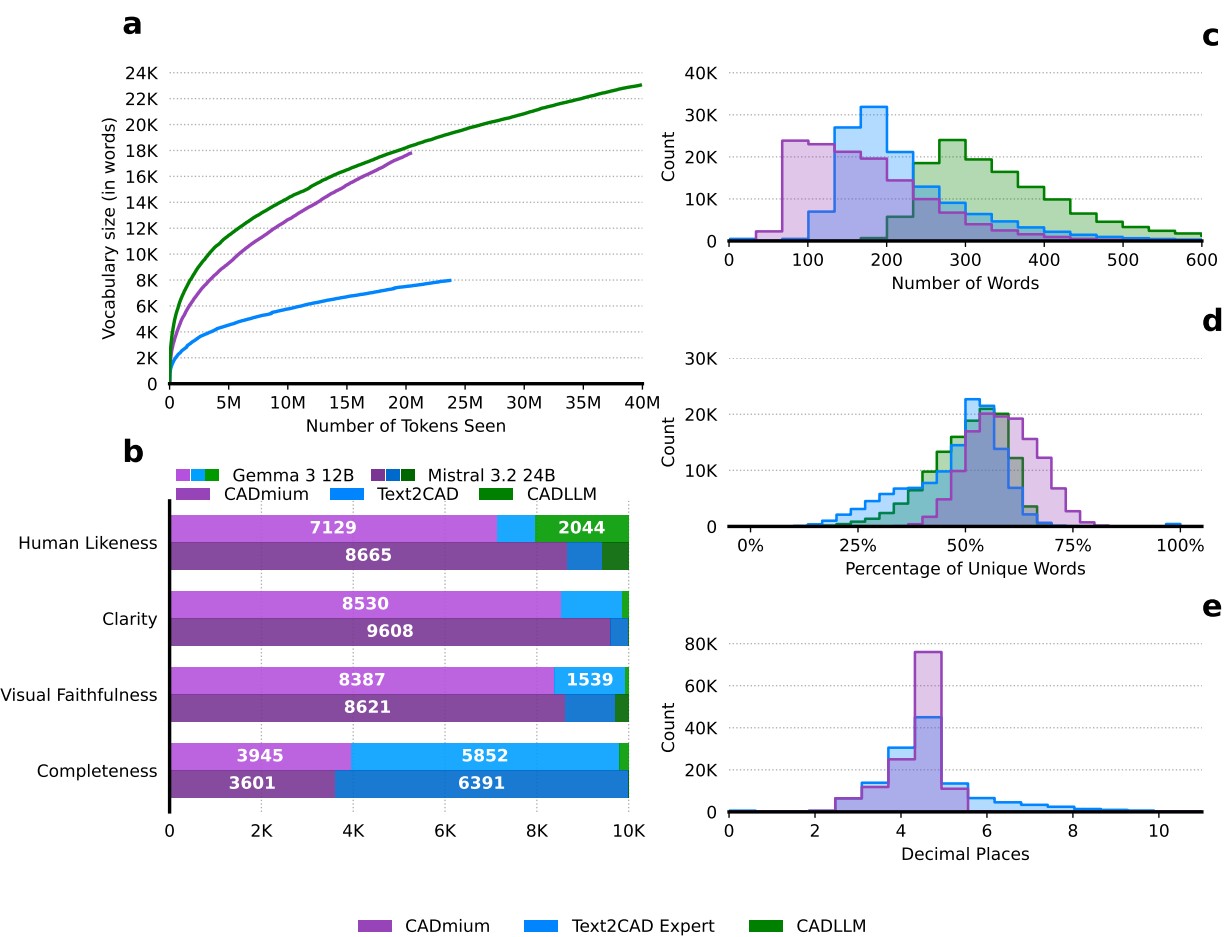

Figure 2: **Comparative analysis of CADmium, CADLLM, and Text2CAD expert-level annotations.** **(a)** Vocabulary growth as a function of token count shows that CADLLM Liao et al. (2025) produces substantially longer annotations, with a vocabulary expansion rate comparable to CADmium and larger than Text2CAD. **(b)** Human-likeness, clarity, visual faithfulness given image renders, and completeness against the minimal JSON as measured by Gemma-3 12B (Team et al., 2025) and Mistral Small 3.2 24B (Mistral, 2025) indicates that CADmium descriptions tend to produce more natural-sounding albeit challenging descriptions. **(c-e)** Distribution of word counts, unique words, and decimal places within numerical expressions per annotation shows that CADmium descriptions are more concise and diverse (**e** reports only CADmium and Text2CAD distributions, as CADLLM annotations contain exclusively integer values).

the excessive precision of Text2CAD, while CADLLM annotations contain exclusively integer values. (Figure 2e). A side-by-side comparison between CADmium and Text2CAD descriptions is provided in Appendix A.

While statistics provide a general feel for the quality of the corpus, they may not tell how individual descriptions compare head-to-head. Therefore, we conducted an LLM-as-a-judge evaluation using Gemma-3 12B (Team et al., 2025) and Mistral Small 3.2 24B (Mistral, 2025) on 10,000 groups of three annotations (one from CADmium, one from Text2CAD, one from CADLLM Liao et al. (2025)). We looked at four key factors: human-likeness, clarity and readability, visual faithfulness to image renders, and completeness with respect to the minimal JSON description. CADmium was preferred for human-likeness and clarity while achieving comparable visual faithfulness. Text2CAD scored higher on completeness, which aligns with our design choice to prioritize natural-sounding concise annotations over the exhaustive, JSON-like detail found in Text2CAD. We argue that such verbosity, while improving completeness, is less representative of how humans naturally describe CAD models and may artificially simplify the task for language models. CADLLM

annotations were typically rated lowest across all metrics, implying their descriptions are less readable and faithful. Notably, CADLLM explicitly separates appearance and parameter descriptions, whereas CADmium and Text2CAD integrate them into a single, interleaved description. The disparity in scores may therefore reflect the LLM judge's preference for this integrated text format over a segmented approach. Details on the evaluation setup and prompts can be found in Appendix C.

For model training and evaluation, we adopted the same train, validation, and test splits as defined by Text2CAD. These splits are derived from filtered samples of the DeepCAD dataset, which were curated to contain approximately 5% cuboid and 5% cylindrical models, reducing their original distribution of roughly 25% cuboid and 8% cylindrical samples in the unfiltered training set. This filtering process resulted in final split sizes of 118,299 for training, 8,925 for validation, and 8,046 for testing.

## 3.2 Model Selection and Training Details

We perform supervised fine-tuning (SFT) on Qwen-2.5 Coder to autoregressively reconstruct CAD histories in minimal JSON format from their textual description. We selected Qwen-2.5 Coder as it is a state-of-the-art instruction-tuned code-LLM, and preliminary experiments with non-code LLMs did not yield promising results. Qwen-2.5 Coder is available in multiple sizes, and we considered four sizes: 1.5B, 3B, 7B, and 14B parameters. We discuss ablations on the model size in Section 4.2.

All models were fine-tuned using AdamW with the default $\beta_1 = 0.9$, $\beta_2 = 0.999$, and $\epsilon = 10^{-8}$. We use a cosine learning rate schedule with 100 warmup steps, reaching a peak learning rate of $2 \times 10^{-4}$, and subsequently decaying to a final rate of 0 at the end of the training. To improve computational efficiency, mixed-precision training with `fp16` was applied alongside Low-Rank Adaptation (LoRA) (Hu et al., 2022) across all linear layers, configured with a rank of 64 and an alpha value of 32.

All experiments were conducted on nodes with four NVIDIA A100 80GB GPUs, utilizing Fully Sharded Data Parallel (FSDP) (Zhao et al., 2023) for efficient distributed training, and achieving an effective batch size of 16. The loss function was computed exclusively on the assistant's generated JSON responses. The models were trained for up to 4 epochs (approximately 29,000 steps), with each complete training run typically lasting between 8 to 12 hours, depending on the specific model size being fine-tuned.

## 3.3 Evaluation

We evaluate all models on unseen 3D objects by generating their CAD sequences using greedy decoding. To assess the quality of the generated outputs, we adopt the following reconstruction and topological metrics from Text2CAD and CAD-MLLM: **Invalidity Ratio** (IR) is the percentage of generated samples that cannot be successfully rendered into valid CAD objects; **F1** is a classification metric computed by aligning generated and ground-truth loops using the Hungarian matching algorithm; **Chamfer Distance** (CD) measures the dissimilarity between generated and reference point clouds; **Segment Error** (SegE) is the scaled absolute difference between the number of segments in the generated and ground-truth samples; **Dangling Edge Length** (DangEL) is the total length of edges that are shared by fewer than two faces; **Self-Intersection Ratio** (SIR) is the proportion of faces that self-intersect; and **Flux Enclosure Error** (FluxEE) quantifies how effectively the mesh encloses a watertight volume.

Additionally, in order to more extensively evaluate the quality of the reconstructed CAD models, we introduce novel metrics designed to capture specific geometric and topological properties. The first metric we introduce is the **Sphericity Discrepancy** (SD) metric. Sphericity quantitatively measures how closely a 3D object resembles a sphere. Given an object with a surface area $s$ and volume $V$, sphericity ($\Psi$) is defined as

$$\Psi = \frac{S}{s} = \frac{\pi^{\frac{1}{3}}(6V)^{\frac{1}{3}}}{s} \tag{1}$$

where $S$ is the surface of a sphere of volume $V$. Given that a sphere achieves the minimum surface area given a particular volume, $\Psi \in (0, 1]$. Sphericity, which is explored in foundational work on particle shape characterization (Wadell, 1932), and used in various applications for quality assessment of 3D parts, is 1 for a perfect sphere and decreases for those that are less compact or have more intricate surfaces relative to

their volume. The final sphericity discrepancy metric quantifies the difference between the predicted object (*pred*) and the ground truth object (*gt*) as the absolute difference of their normalized ratios:

$$\text{SD} = |\Psi_{pred} - \Psi_{gt}| \tag{2}$$

A lower SD value indicates greater similarity in the overall compactness between the predicted and ground truth models.

The second metric we introduce, termed the **Discrete Mean Curvature Difference** (DMCD), quantifies local dissimilarities in surface geometry. To achieve this, we compute a discrete mean curvature value at each vertex for both the predicted and ground truth meshes. This computation follows the principles for discrete curvature estimation presented by Cohen-Steiner & Morvan (2003), leveraging an implementation available in the `trimesh` library (Dawson-Haggerty et al.). Specifically, for each vertex, the discrete mean curvature value $\kappa$ is calculated as the sum of the signed dihedral angles of all mesh edges contained within a sphere of a defined radius $r$ centered at that vertex. This per-vertex value $\kappa$ is inherently dimensionless. The curvature at each vertex is then averaged, resulting in $\bar{\kappa}$. We compute the absolute difference between the average $\kappa$ values of the predicted and ground truth meshes:

$$\text{DMCD}_{\text{vertex}} = |\bar{\kappa}_{pred} - \bar{\kappa}_{gt}| \tag{3}$$

As $\bar{\kappa}$ quantifies the global measure of curvature over the vertices of the mesh, a lower DMCD value indicates a greater similarity in the surface geometry between the predicted and ground truth models.

We also introduce the **Exact Euler Characteristic Match** (EECM) metric. The Euler characteristic, denoted as $\chi$, is a well-established topological invariant that describes a shape's structure regardless of how it is bent or deformed. For a 3D manifold mesh, it is commonly computed using the formula $\chi = V - E + F$, where $V$ is the number of vertices, $E$ the number of edges, and $F$ the number of faces (Richeson, 2012). The EECM is a binary metric defined as

$$\text{EECM} = \begin{cases} 1 & \text{if } \chi_{pred} = \chi_{gt} \\ 0 & \text{otherwise} \end{cases} \tag{4}$$

This metric serves as a basic check for whether the generated model preserves the essential topological features, like the number of holes or separate shells of the target object.

A fundamental quality indicator for generated 3D shapes, especially those intended for physical simulation, manufacturing, or volumetric analysis, is whether they constitute a **Watertight Mesh**. A watertight mesh is a closed, 2-manifold polygonal surface that clearly separates an interior volume from an exterior space, meaning it has no holes, boundary edges, or non-manifold configurations that would prevent the unambiguous definition of its volume (Botsch et al., 2010). We report the percentage of generated CAD models that are watertight as a measure of their geometric validity and usability. In our evaluation pipeline, we compute SD, EECM, and DMCD only if both the predicted and ground truth meshes are watertight.

## 4 Experiments

This section provides an overview of our experimental setup and presents the evaluation of the proposed CAD generation pipeline. We conduct experiments across four datasets: the original Text2CAD dataset (Khan et al., 2024), our re-annotated DeepCAD dataset (CADmium), the *CADPrompt* dataset (Alrashedy et al., 2024) containing 200 expert-annotated objects from DeepCAD (Wu et al., 2021), and the Fusion360 Reconstruction dataset (Willis et al., 2021) annotated using our pipeline.

A key aspect of our evaluation protocol is that we must ensure that generated 3D models meet specific shape and structural requirements to correctly compute metrics. First, all metrics require that the generated JSON, or vector representation in the case of Text2CAD, successfully convert into a valid 3D mesh. The proportion of samples failing this basic requirement is quantified by the invalidity ratio (IR). Beyond this universal prerequisite, EECM, DMCD, and SD additionally require both predicted and ground-truth meshes to be watertight.

Due to these varying requirements, the number of objects that can correctly be evaluated per metric differs across experiments, for instance when comparing different model sizes. To maximize coverage, we report results using the entire subset of valid samples per metric and model, and include the corresponding sample size. However, as these subsets vary from one model to the next, it may hinder direct comparison. To address this, we include a complementary analysis in Appendix B, where all metrics are recomputed on the intersection of samples that satisfy all relevant conditions across compared experiments, ensuring fair and consistent comparisons.

Additionally, for the Chamfer Distance, we observed that rare outliers with extremely high values skewed the mean, rendering it less informative. We therefore report the median CD as a more robust alternative. All metrics are accompanied by their 95% confidence interval computed over individual test samples for a single model run.

To assess whether observed differences between models are statistically significant, we apply different hypothesis tests depending on the evaluation protocol. When comparing results on non-identical sets of valid meshes (main paper tables), we treat the samples as independent and use unpaired tests: Welch's t-test if both groups appear approximately normal, otherwise the Mann–Whitney U test, and for binary outcomes such as validity and watertightness we use Fisher's exact test or the Chi-square test with Yates' correction when expected counts are sufficient. When comparing results on the common subset of objects valid across models (Appendix B), we use paired tests: the Wilcoxon signed-rank test for continuous metrics and Mc-Nemar's exact test for binary outcomes. In all cases, we report differences as statistically significant if the adjusted p-value (Benjamini–Hochberg correction within each metric) is below 0.05.

## 4.1 Generalization to Human-Annotated Prompts (*CADPrompt*)

To assess our models' ability to generalize to prompts authored by humans, we evaluated them on the *CADPrompt* dataset (Alrashedy et al., 2024). This dataset comprises expert-written natural language instructions for 200 DeepCAD samples, serving as a crucial test for out-of-distribution generalization. Table 1 presents the results, comparing our Qwen Coder 14B model trained on CADmium data against the original Text2CAD model as a key baseline. A corresponding common subset analysis is provided in Appendix B.

The Text2CAD model achieves a remarkably low Invalidity Ratio, a characteristic of its specialized vector representation for CAD sequences, which is designed with a strong inductive bias that guides the model to generate valid outputs. When comparing our Qwen Coder 14B model trained on CADmium annotations against this Text2CAD model baseline, our LLM-based approach proves to be a competitive alternative, by demonstrating substantially better Line, Arc, Circle, Extrusion F1 scores, SD, and CD. The Text2CAD model excels in overall geometric fidelity as measured by SIR, and achieves higher Watertightness. This highlights that while the Text2CAD model's design ensures high validity and strong performance on some global shape metrics, our LLM approach offers significant advantages in understanding instructions for specific, geometric elements and structural properties.

## 4.2 Impact of Model Scale

We investigated the influence of model scale by fine-tuning Qwen-2.5 Coder models of 1.5B, 3B, 7B, and 14B parameters on our CADmium training data. These models were evaluated on both the CADmium test set and the Fusion360 Reconstruction dataset to assess performance scalability and generalization. Detailed metrics are presented in Table 2 and Table 3, with a corresponding analysis on a common subset of valid samples provided in Appendix B.

A primary benefit of increasing model size is the consistent reduction in the Invalidity Ratio (IR). On the CADmium test set, IR drops progressively with larger models, and the 14B model achieves a notably lower IR on the more challenging Fusion360 dataset as well. Accuracy in reconstructing geometric primitives, as measured by F1 scores, generally improves or remains robust with increasing model size, particularly when compared on a common subset of generated objects (see Appendix B). Metrics related to fine-grained mesh quality, such as DangEL and SD, also tend to show improvements with larger models, suggesting better

Table 1: Performance comparison on the human-annotated *CADPrompt* dataset. Metrics are calculated on the set of valid meshes generated by each model individually. Reported values include 95% confidence intervals in brackets. Boldface indicates the better-performing model when the difference is statistically significant at $p < 0.05$. Boldface indicates statistically significant differences ($p < 0.05$); details of the statistical testing procedure are provided in Section 4.

| Model
Trained on | Text2CAD
Text2CAD | Qwen-2.5 Coder 14B
CADmium |
|---|---|---|
| IR (%) ↓ | **1.57** | 6.28 |
| Num Valid | 188 | 179 |
| Line F1 ↑ | 0.67 [0.62, 0.71] | 0.69 [0.64, 0.74] |
| Arc F1 ↑ | 0.00 [0.00, 0.00] | **0.22** [0.11, 0.35] |
| Circle F1 ↑ | 0.31 [0.23, 0.41] | **0.61** [0.51, 0.71] |
| Extrusion F1 ↑ | 0.84 [0.82, 0.87] | **0.89** [0.87, 0.92] |
| CD mean ↓ | 221.42 [185.79, 258.88] | 205.46 [175.29, 237.00] |
| CD median ↓ | 127.70 [87.59, 162.91] | 116.75 [81.64, 179.09] |
| SIR ↓ | **0.02** [0.01, 0.04] | 0.11 [0.07, 0.15] |
| DangEL ↓ | **0.20** [0.03, 0.44] | 2.98 [1.67, 4.55] |
| SegE ↓ | **0.09** [0.03, 0.17] | 0.78 [0.48, 1.14] |
| FluxEE ($\times 10^2$) ↓ | **0.04** [0.00, 0.12] | 2.24 [1.01, 3.73] |
| Watertightness (%) ↑ | **95.74** [92.55, 98.40] | 83.80 [78.21, 88.83] |
| Num Watertight | 165 | 136 |
| EECM ↑ | 0.66 [0.59, 0.73] | 0.68 [0.60, 0.75] |
| DMCD ($\times 10^3$) ↓ | 13.39 [11.61, 15.26] | **8.42** [6.63, 10.35] |
| SD mean ($\times 10^2$) ↓ | 14.78 [12.74, 17.02] | **7.02** [5.05, 9.26] |
| SD median ($\times 10^2$) ↓ | 11.15 [8.68, 13.64] | **1.54** [1.07, 2.15] |

Table 2: Impact of Qwen Coder model scale (1.5B, 3B, 7B, 14B parameters), trained on CADmium annotations, evaluated on the **CADmium test set**. Metrics are calculated on the set of valid meshes generated by each model individually for each dataset. Reported values include 95% confidence intervals in brackets. Statistical significance is assessed with statistical tests on adjacent sizes (1.5B→3B, 3B→7B, 7B→14B). Bold indicates a size that is significantly better than the next smaller size at $p < 0.05$. Details of the statistical testing procedure are provided in Section 4.

| Metric | 1.5B | 3B | 7B | 14B |
|---|---|---|---|---|
| IR (%) ↓ | 11.86 | 8.51 | 5.93 | **3.33** |
| Num Valid | 7092 | 7361 | 7569 | 7778 |
| Line F1 ↑ | 0.90 [0.90, 0.91] | **0.91** [0.90, 0.91] | 0.91 [0.90, 0.91] | **0.92** [0.91, 0.92] |
| Arc F1 ↑ | 0.70 [0.68, 0.73] | **0.74** [0.72, 0.76] | 0.73 [0.71, 0.75] | **0.78** [0.76, 0.80] |
| Circle F1 ↑ | 0.87 [0.86, 0.88] | **0.91** [0.90, 0.92] | 0.89 [0.89, 0.90] | **0.92** [0.91, 0.92] |
| Extrusion F1 ↑ | 0.98 [0.98, 0.98] | **0.98** [0.98, 0.98] | 0.98 [0.98, 0.99] | 0.99 [0.98, 0.99] |
| CD mean ↓ | 70.48 [66.65, 74.52] | **43.91** [40.52, 47.36] | 63.11 [59.41, 66.91] | **42.69** [39.71, 45.85] |
| CD median ↓ | 0.31 [0.29, 0.32] | 0.24 [0.23, 0.25] | 0.27 [0.26, 0.28] | 0.23 [0.22, 0.25] |
| SIR ↓ | 0.04 [0.04, 0.04] | 0.05 [0.04, 0.05] | 0.04 [0.04, 0.05] | 0.04 [0.04, 0.04] |
| DangEL ↓ | 1.18 [1.03, 1.35] | 1.35 [1.20, 1.50] | 1.10 [0.98, 1.23] | 1.13 [1.00, 1.27] |
| SegE ↓ | 0.48 [0.42, 0.55] | 0.52 [0.46, 0.57] | 0.48 [0.43, 0.53] | 0.54 [0.45, 0.67] |
| FluxEE ($\times 10^2$) ↓ | 0.38 [0.30, 0.47] | 0.49 [0.39, 0.60] | **0.36** [0.29, 0.43] | 0.39 [0.31, 0.48] |
| Watertightness (%) ↑ | 90.38 [89.71, 91.05] | 90.31 [89.63, 90.99] | 90.54 [89.88, 91.20] | 90.78 [90.13, 91.42] |
| Num Watertight | 6205 | 6408 | 6575 | 6779 |
| EECM ↑ | 0.87 [0.86, 0.88] | **0.89** [0.88, 0.90] | 0.88 [0.87, 0.89] | 0.89 [0.88, 0.90] |
| DMCD ($\times 10^3$) ↓ | 2.97 [2.78, 3.16] | **2.68** [2.49, 2.87] | 2.73 [2.54, 2.93] | **2.67** [2.48, 2.86] |
| SD mean ($\times 10^2$) ↓ | 2.90 [2.70, 3.11] | **2.19** [2.02, 2.37] | 2.23 [2.06, 2.41] | **1.99** [1.83, 2.15] |
| SD median ($\times 10^2$) ↓ | 0.00 [0.00, 0.00] | 0.00 [0.00, 0.00] | 0.00 [0.00, 0.00] | 0.00 [0.00, 0.00] |

adherence to manifold properties. Some examples of the reconstructions generated by the 14B model are shown in Figure 3.

Table 3: Impact of Qwen Coder model scale (1.5B, 3B, 7B, 14B parameters), trained on CADmium annotations, evaluated on the **Fusion360 test set**. Metrics are calculated on the set of valid meshes generated by each model individually for each dataset. Reported values include 95% confidence intervals in brackets. Statistical significance is assessed with statistical tests on adjacent sizes (1.5B→3B, 3B→7B, 7B→14B). Bold indicates a size that is significantly better than the next smaller size at $p < 0.05$. Details of the statistical testing procedure are provided in Section 4.

| Metric | 1.5B | 3B | 7B | 14B |
|---|---|---|---|---|
| IR (%) ↓ | 22.62 | 23.38 | 16.79 | **12.58** |
| Num Valid | 6428 | 6365 | 6912 | 7262 |
| Line F1 ↑ | 0.79 [0.78, 0.80] | **0.83** [0.83, 0.84] | 0.83 [0.82, 0.83] | **0.84** [0.83, 0.85] |
| Arc F1 ↑ | 0.63 [0.62, 0.65] | **0.70** [0.68, 0.71] | 0.67 [0.65, 0.68] | **0.71** [0.70, 0.73] |
| Circle F1 ↑ | 0.84 [0.83, 0.85] | **0.90** [0.89, 0.90] | 0.88 [0.87, 0.88] | **0.90** [0.89, 0.90] |
| Extrusion F1 ↑ | 0.97 [0.97, 0.98] | **0.98** [0.97, 0.98] | 0.98 [0.98, 0.98] | **0.98** [0.98, 0.98] |
| CD mean ↓ | 226.46 [218.86, 234.38] | **191.94** [184.22, 199.68] | 206.39 [199.19, 213.78] | **202.00** [194.52, 209.37] |
| CD median ↓ | 110.18 [104.45, 117.15] | **76.34** [69.24, 83.42] | 92.07 [86.51, 97.20] | **76.05** [70.08, 82.17] |
| SIR ↓ | 0.05 [0.04, 0.05] | 0.05 [0.05, 0.06] | 0.05 [0.05, 0.06] | **0.04** [0.04, 0.05] |
| DangEL ↓ | 1.36 [1.23, 1.51] | 1.41 [1.27, 1.56] | 1.54 [1.40, 1.70] | **1.26** [1.12, 1.42] |
| SegE ↓ | 0.59 [0.54, 0.65] | **0.60** [0.55, 0.67] | 0.62 [0.56, 0.67] | 0.65 [0.57, 0.75] |
| FluxEE ($\times 10^2$) ↓ | 0.28 [0.21, 0.38] | 0.31 [0.25, 0.38] | 0.30 [0.23, 0.38] | 0.38 [0.24, 0.61] |
| Watertightness (%) ↑ | 86.62 [85.80, 87.48] | 86.50 [85.67, 87.35] | 85.21 [84.38, 86.05] | **87.35** [86.57, 88.09] |
| Num Watertight | 5241 | 5192 | 5504 | 5941 |
| EECM ↑ | 0.77 [0.76, 0.78] | **0.81** [0.80, 0.82] | 0.78 [0.77, 0.79] | **0.80** [0.79, 0.81] |
| DMCD ($\times 10^3$) ↓ | 5.19 [4.91, 5.48] | **4.09** [3.84, 4.35] | 4.90 [4.62, 5.19] | **4.43** [4.18, 4.69] |
| SD mean ($\times 10^2$) ↓ | 4.53 [4.25, 4.81] | **3.18** [2.96, 3.41] | 3.87 [3.63, 4.12] | **3.45** [3.23, 3.68] |
| SD median ($\times 10^2$) ↓ | 0.00 [0.00, 0.00] | 0.00 [0.00, 0.00] | 0.00 [0.00, 0.00] | 0.00 [0.00, 0.00] |

### 4.3 Impact of Annotation Quality and Training/Evaluation Congruence

To isolate the impact of annotation quality, we adopted the original Text2CAD modeling approach, which utilizes a custom transformer architecture and a specialized vector representation for CAD sequences. We trained this Text2CAD architecture separately on our CADmium dataset and the original Text2CAD dataset, followed by cross-set evaluations. The results are presented in Table 4. An analysis on a common subset of samples is available in Appendix B.

Our primary finding is that when models are trained and evaluated in-distribution (i.e., on datasets with the same annotation style), the Text2CAD architecture trained on our CADmium annotations achieves performance that is broadly comparable or superior to the same architecture trained on the original Text2CAD annotations. As detailed in Table 3, the CADmium-trained model (when evaluated on CADmium data) shows lower IR, higher F1 scores for all primitive types, better EECM, DMCD, and SD compared to the Text2CAD-trained model evaluated on Text2CAD data. This is a significant result, considering that CADmium's annotations are designed to be more human-like and are therefore less directly aligned with the ground truth JSON structure than Text2CAD's more template-driven expert prompts.

Conversely, both models exhibit a performance degradation when evaluated out-of-distribution. The model trained on CADmium annotations sees a notable drop in F1 scores and an increase in CD when tested on the Text2CAD dataset. Similarly, the model trained on Text2CAD annotations shows considerably worse geometric fidelity (e.g., higher CD) and higher IR when evaluated on the CADmium dataset's more natural prompts.

## 5 Limitations

While CADmium demonstrates the efficacy of fine-tuning code-LLMs for text-to-CAD generation, multiple limitations remain.

A primary limitation is the absence of large-scale human evaluation for the generated annotations. We employed an LLM-as-a-judge paradigm to scale our assessment, but this approach may carry intrinsic biases.

Table 4: Impact of annotation quality using the Text2CAD architecture. Models were trained on either CADmium or Text2CAD annotations and evaluated on both the CADmium and Text2CAD test sets. Metrics are calculated on the set of valid meshes generated by each model–dataset configuration individually. Reported values include 95% confidence intervals in brackets. Boldface indicates statistically significant differences ($p < 0.05$); details of the statistical testing procedure are provided in Section 4.

| Evaluated on | CADmium | | Text2CAD | |
|---|---|---|---|---|
| Trained on | CADmium | Text2CAD | CADmium | Text2CAD |
| IR (%) ↓ | **3.07** | 4.44 | 4.95 | **4.34** |
| Num Valid | 7799 | 7447 | 7648 | 7697 |
| Line F1 ↑ | **0.88** [0.87, 0.88] | 0.73 [0.72, 0.74] | 0.64 [0.63, 0.64] | **0.82** [0.81, 0.83] |
| Arc F1 ↑ | **0.58** [0.56, 0.60] | 0.21 [0.19, 0.23] | 0.24 [0.23, 0.26] | **0.38** [0.36, 0.40] |
| Circle F1 ↑ | **0.87** [0.86, 0.88] | 0.57 [0.56, 0.59] | 0.47 [0.46, 0.48] | **0.76** [0.74, 0.77] |
| Extrusion F1 ↑ | **0.98** [0.98, 0.98] | 0.89 [0.89, 0.90] | 0.81 [0.80, 0.81] | **0.93** [0.93, 0.94] |
| CD mean ↓ | **53.55** [50.17, 57.06] | 145.87 [141.11, 150.64] | 107.99 [104.16, 111.93] | **29.38** [27.50, 31.35] |
| CD median ↓ | **0.33** [0.32, 0.36] | 51.67 [47.72, 56.13] | 34.23 [31.87, 36.29] | **0.38** [0.35, 0.42] |
| SIR ↓ | 0.05 [0.05, 0.05] | 0.05 [0.04, 0.05] | 0.08 [0.08, 0.09] | **0.05** [0.04, 0.05] |
| DangEL ↓ | 1.66 [1.50, 1.83] | **1.16** [1.02, 1.31] | 2.56 [2.34, 2.78] | **1.13** [1.01, 1.26] |
| SegE ↓ | 0.80 [0.68, 0.95] | **0.53** [0.47, 0.61] | 1.17 [1.08, 1.28] | **0.55** [0.49, 0.62] |
| FluxEE ($\times 10^2$) ↓ | **0.46** [0.39, 0.55] | 0.59 [0.46, 0.73] | 0.97 [0.84, 1.10] | **0.46** [0.36, 0.57] |
| Watertightness (%) ↑ | 87.63 [86.90, 88.36] | **91.90** [91.29, 92.50] | 82.48 [81.63, 83.33] | **90.52** [89.85, 91.17] |
| Num Watertight | 6562 | 6555 | 5957 | 6644 |
| EECM ↑ | **0.86** [0.85, 0.87] | 0.76 [0.75, 0.77] | 0.66 [0.65, 0.67] | **0.82** [0.81, 0.83] |
| DMCD ($\times 10^3$) ↓ | **3.65** [3.44, 3.87] | 8.29 [7.98, 8.62] | 9.41 [9.11, 9.73] | **5.35** [5.10, 5.61] |
| SD mean ($\times 10^2$) ↓ | **2.71** [2.54, 2.87] | 8.69 [8.39, 8.99] | 11.52 [11.14, 11.91] | **5.04** [4.79, 5.29] |
| SD median ($\times 10^2$) ↓ | **0.00** [0.00, 0.00] | 3.65 [3.41, 4.01] | 5.33 [4.90, 5.69] | **0.11** [0.06, 0.16] |

LLM judges can exhibit preference leakage, favoring outputs that resemble their own training data or specific stylistic patterns, potentially preferring machine-generated text. To mitigate this evaluation bias, we include two families of LLMs, Gemma-3 12B and Mistral Small 3.2 24B, for judging and comparing our annotations with previous datasets.

Our dataset focuses on expert-level, geometrically precise descriptions to ensure unambiguous reconstruction. However, this may limit the model's ability to generalize to abstract, vague, or purely functional descriptions often used by non-experts (e.g., "design a handle for a mug"). The model may struggle when instructions lack explicit geometric parameters. Further, the current scope is limited to sketch and extrude operations. Advanced CAD features such as chamfers and fillets operations are not currently supported, limiting the complexity of manufacturable objects. However, unlike the rigid structure of vectorized representations, text-based CAD sequences are inherently flexible, facilitating the addition of novel operations.

# 6 Discussion and Conclusion

Despite the current progress of foundation models that have multimodal capabilities, achieving state-of-the-art in language, image, and 3D object generation (Xiang et al., 2024) and understanding, there is still plenty of room for improvement when it comes to domain-specific tasks. Much of this improvement depends on the availability of domain-relevant data and the capacity of large machine learning models to learn from it. Fine-tuning large language models for generating application-specific 3D structural entities, such as small molecules, proteins, and materials, is an active area of research that has shown remarkable progress in recent years. Likewise, CAD also deals with 3D mechanical objects, and automating this process can rapidly advance manufacturing sectors and 3D printing. Specifically, text-conditioned CAD model generation and completion can greatly help engineers and designers to quickly optimize the design process with simple natural language instructions.

With the absence of large and good-quality human-annotated textual descriptions of CAD objects that are publicly available, it is necessary to utilize powerful and multimodal language models that can generate human-level annotations given the images and design history of CAD objects. In this regard our main

goals were to further the state-of-the-art in terms of the quality of the text annotations, improve the training process by using instruction-tuned Code Qwen models, and augment the evaluation pipeline by including additional metrics that can provide deeper structural and topological insights (Appendix D). We also highlight some important limitations in our work. Although we demonstrated that GPT-generated prompts exhibit human-like characteristics, further investigation is needed to evaluate their generalization to diverse types of human prompts. Additionally, we observed a few outlier reconstructions with high Chamfer distances from the ground truths, which may be mitigated by increasing the training data and model size. In future work, we plan to extend our approach with a multimodal framework that facilitates editing CAD designs. Overall, we wish to integrate some of the advancements from recent concurrent works, particularly those focused on multimodality and enhanced controllability with CADmium's training methodology.

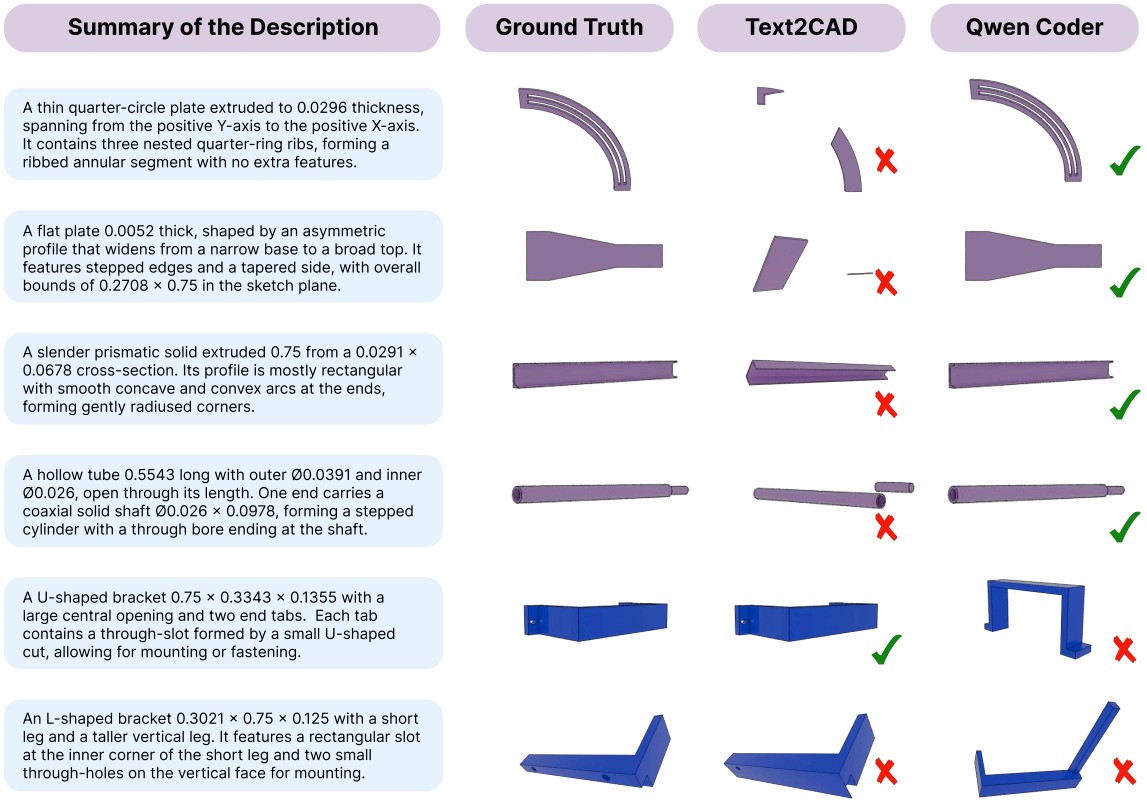

Figure 3: **Examples of generated objects from the CADmium test set:** 3D models are generated by both Qwen2.5 Coder (fine-tuned) and Text2CAD (trained) on the CADmium training set. The examples are shown using CADmium's expert prompts, with the natural-language descriptions shortened for readability. We present both successful and failed reconstructions from Qwen2.5 Coder and compare them against the outputs of Text2CAD.

## 7   Acknowledgments

Sarath Chandar is supported by the Canada CIFAR AI Chairs program, the Canada Research Chair in Lifelong Machine Learning, and the NSERC Discovery Grant. This research was enabled in part by compute resources provided by Mila and Compute Canada. This work was supported by funding from Ansys.

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

## A   Example Annotations

This appendix provides illustrative examples comparing our CADmium annotations, generated using GPT-4.1, with the expert-level annotations from the Text2CAD dataset for the same CAD models. These qualitative comparisons, presented in Figures 4 through 8, highlight several improvements discussed in Section 3.1 and quantitatively supported by the analysis in Figure 2.

The selected examples specifically demonstrate instances where:

- CADmium annotations offer more natural, human-like phrasing, avoiding the overly programmatic or JSON-key-like references sometimes found in Text2CAD expert annotations (as shown in Figure 4).

- Our pipeline's use of a more powerful vision-language model (GPT-4.1) leads to more accurate capture of geometric details, such as correctly identifying the number of holes in an object, compared to potential misinterpretations in Text2CAD annotations (Figure 5).

- CADmium annotations can leverage broader knowledge to describe objects by their resemblance to known items or by their overall form (e.g., "a clamp-like object"), providing a more holistic understanding than descriptions that merely summarize CAD construction entities, as sometimes seen in Text2CAD (Figure 6).

- Our annotations provide clear, correct, and actionable geometric information, in contrast to some Text2CAD examples that may be vague, contain errors, or lack sufficient detail for unambiguous reconstruction (Figure 7).

- CADmium annotations represent numerical values with a natural level of precision, unlike the excessively long decimal expansions often present in Text2CAD annotations, making our prompts more aligned with human expression (Figure 8).

Each figure typically presents the CADmium annotation alongside the corresponding Text2CAD expert annotation, often accompanied by visual renders of the 3D object for better context and comparison.



**Text2CAD Expert**
Create the first part of the CAD model, a cylindrical object with a hollow center. Begin by setting up a new coordinate system with Euler angles of (0.0, 0.0, 0.0) and a translation vector of (0.0, 0.0, 0.0).
For part 1, create a 2D sketch on the X-Y plane of this coordinate system. The sketch consists of `face_1`, which has two loops (`loop_1` and `loop_2`). `Loop_1` contains a single circle named `circle_1` with a center at (0.2083, 0.2083) and a radius of 0.2083.
`Loop_2` contains three curves: a line (`line_1`), a line (`line_2`) and an arc (`arc_1`). The first curve, `line_1`, starts at (0.1383, 0.1176) and ends at (0.1994, 0.1176). The second curve, `line_2`, starts at (0.1994, 0.1176) and ends at (0.2783, 0.1176). The third curve, `arc_1`, starts at (0.2783, 0.1176), has a midpoint at (0.2083, 0.3229), and ends at (0.1383, 0.1176).
Next, scale the 2D sketch with a scaling factor of 0.4167. Then, extrude the scaled 2D sketch by `0.3333` units in the positive Z direction and `0.0` units in the negative Z direction. Create a new solid body for this operation and complete the first part of the CAD model.
The first part of the CAD model has the following dimensions: for length and width, 0.41666666666666663, and for height, 0.3333333333333333.

**CADmium**
Begin by creating a solid cylinder with a diameter of 0.4167 and a height of 0.3333. On the top face of the cylinder, sketch a circle centered at (0.2083, 0.2083) with a radius of 0.2083. Inside this circle, define a closed loop comprised of two horizontal lines and an arc: draw the first line from (0.1383, 0.1176) to (0.1994, 0.1176), the second line from (0.1994, 0.1176) to (0.2783, 0.1176), and connect the end points of these lines back to the starting point with an arc that has its midpoint at (0.2083, 0.3229), forming a segment that cuts across the base of the circle. Extrude this internal profile fully through the cylinder to create a cylindrical hole with a flat chord across its lower portion as defined by the lines and arc, resulting in a cylinder with a partial circular cutout running through its center.

Figure 4: Annotation comparison: CADmium (light purple) uses natural language, while Text2CAD (light blue) references JSON-like keys (e.g., 'face_1').

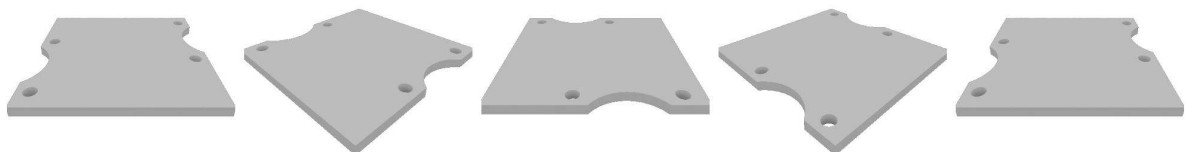

**Text2CAD Expert**
Create the first part of the CAD model, a rectangular metal plate with rounded corners and **two holes** on the top side. Begin by setting up a new coordinate system with Euler angles of (0.0, 0.0, 0.0) and a translation vector of (0.0, 0.0, 0.0).
Next, draw a 2D sketch on the X-Y plane of the coordinate system, and create a rectangular outline consisting of straight lines and arcs as provided in the detailed instructions.
Create **two circular holes** at the specified locations: the first circle has a center at (0.0781, 0.6771) and a radius of 0.026; the second circle has a center at (0.2969, 0.0521) and a radius of 0.026. Ensure that all loops are closed and then scale the sketch by 0.75 using the sketch\\_scale parameter.
Transform the scaled 2D sketch into a 3D sketch by applying the coordinate system's orientation (euler angles) and translation vector. Extrude the sketch by 0.0391 to create a 3D solid model representing part 1.
The part has the following dimensions:
- Height: 0.03906249999999999
- Width: 0.7499999999999998
- Length: 0.7499999999999998

**CADmium**
Begin by sketching a rectangle measuring 0.75 in length and 0.7292 in width. On the lower edge (along the length), create a vertical line from the origin (0.0, 0.0) to (0.3499, 0.0). Add a convex arc from (0.3499, 0.0) through mid-point (0.4844, 0.0664) to (0.6188, 0.0), then continue the bottom edge with a horizontal line to (0.75, 0.0).
From (0.75, 0.0), draw a vertical line upwards to (0.75, 0.7292), forming the right edge. On the top edge, draw a horizontal line from (0.75, 0.7292) to (0.4001, 0.7292). From there, add a concave arc from (0.4001, 0.7292), passing through (0.2656, 0.6628) as the mid-point, and ending at (0.1312, 0.7292). Complete the rectangle by drawing a horizontal line from (0.1312, 0.7292) to (0.0, 0.7292), and a vertical line down to the origin (0.0, 0.0).
Add **four through-holes** with a diameter of 0.052 (radius 0.026), located at the following centers: (0.0781, 0.6771), (0.2969, 0.0521), (0.4531, 0.6771), and (0.6719, 0.0521).
Extrude the entire profile perpendicular to the sketch plane to a thickness of 0.0391.

Figure 5: Geometric detail capture: CADmium (light purple) correctly identifies four holes, unlike Text2CAD (light blue) which describes two, highlighting improved visual understanding.

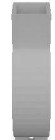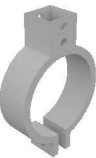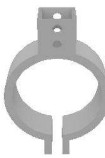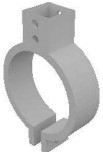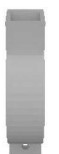

**Text2CAD Expert**
Create the first part of the CAD model: a rectangular prism with a cylindrical hole in its center. Begin by setting up a new coordinate system with Euler angles of (0.0, 0.0, -90.0) and a translation vector of (0.0002, 0.171, 0.0).
Create the first face of the sketch with two loops. **Loop 1 is composed of 11 curves and 3 arcs, as well as 3 circles**, the first of which has a center at (0.2577, 0.6338) and a radius of 0.0202. The second loop in this face is a circle with a center at (0.2577, 0.7086) and a radius of 0.0202.
Extrude the sketch along the normal direction by a depth of 0.171, resulting in the first part of the CAD model. Ensure that the extrusion depth towards the normal is 0.171 and opposite the normal is 0.0. Scale the sketch by 0.75 and create a new solid body for this operation.
The first part of the CAD model has the following dimensions: length of 0.1394, width of 0.1394, and height of 0.171 units.

**CADmium**
Begin by sketching the side profile of a **clamp-like ring** on the base plane, starting with three major arcs and several straight segments. Draw an initial arc from approximately (0.1728, 0.5901) through (0.0, 0.3357) to (0.1936, 0.0967). From there, continue with a line straight down to (0.1936, 0.0002), right to (0.2294, 0.0002), up to (0.2294, 0.0902) and then further up to (0.2294, 0.1295). Next, add a convex arc from (0.2294, 0.1295) through (0.2579, 0.5654) to (0.2856, 0.1294). Draw a short downward line to (0.2856, 0.0901), then further down to (0.2856, 0.0). Take a horizontal line to (0.3236, 0.0), up to (0.3236, 0.0972), and create a third arc from (0.3236, 0.0972) through (0.5154, 0.3359) to (0.3438, 0.5896). Draw a vertical line upward to (0.3438, 0.75), horizontally left to (0.1728, 0.75), and finally straight down to close at (0.1728, 0.5901).
Within the upper region, add two through-holes as circles: one centered at (0.2577, 0.6338) and another at (0.2577, 0.7086), each with a radius of 0.0202. Extrude this profile symmetrically to a thickness of 0.171, forming the main body of the clamp.
On the side of the clamp near the split section, make a cut by sketching a circle with a radius of 0.0202, centered at (0.0202, 0.0202) on the relevant face. Cut this feature through the body for a total depth of 0.3542, with the cut axis perpendicular to the ring surface.
At the top of the structure (positioned at the maximum height of the clamp), add a rectangular slot by sketching a square of 0.1394 by 0.1394, with one corner at (0.0, 0.0). Cut the slot downward into the ring for a depth of 0.1534, forming a mounting or attachment feature that aligns with the existing through-holes.
Ensure the internal curves, outer profile, and slots align concentrically and orthogonally on their respective faces, preserving the clamp's open, split-ring geometry and the structural attachment at the top.

Figure 6: Descriptive abstraction: CADmium (light purple) identifies a "clamp-like ring", while Text2CAD (light blue) summarizes CAD entities with less descriptive power.

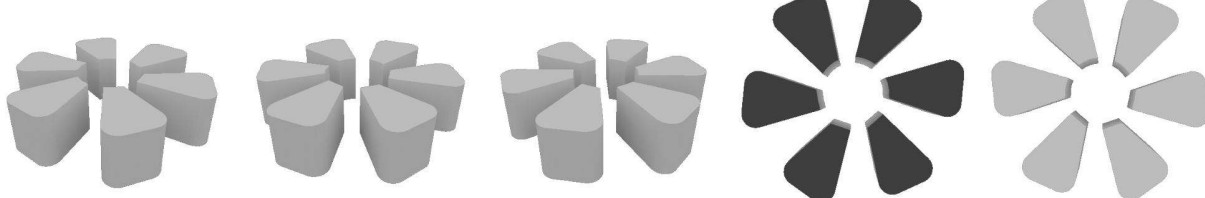

**Text2CAD Expert**
Create the first part of the CAD model, a rectangular prism with a curved top. Begin by setting up a new coordinate system with Euler angles of (0.0, 0.0, 0.0) and a translation vector of (0.0002, 0.0029, 0.0).

Next, create six sketches on the X-Y plane of the coordinate system.

- Face 1: **Draw a loop consisting of lines and arcs defined by specific points**.
- Face 2: Draw a loop consisting of arcs and lines defined by specific points.
- Face 3: Draw a loop consisting of lines and arcs defined by specific points.
- Face 4: Draw a loop consisting of lines and arcs defined by specific points.
- Face 5: Draw a loop consisting of arcs and lines defined by specific points.
- Face 6: Draw a loop consisting of lines and arcs defined by specific points.

**CADmium**
Create a pattern composed of **six identical solid bodies**, each shaped like a rounded wedge, arranged evenly in a radial pattern around a central axis with a small gap between each segment. Model a single segment first: Begin by sketching the outline of the wedge on a 2D plane. The outer curved profile is defined by a combination of straight lines and arcs: start with a short straight section, followed by a convex arc sweeping smoothly to a longer straight side, transitioning to another tangent arc, and completing the profile with a final line that returns to the starting point, enclosing the form. The cross-sectional profile extends to a total length of approximately 0.75 and a maximum width of about 0.72 at its widest point.

Extrude this closed sketch symmetrically upward to a height of 0.19, yielding a 3D body with smoothly blended edges. After the first segment is created, pattern this identical body five more times radially, rotating each one by 60 degrees around the central vertical axis, so that all six segments form a near-complete flower or propeller-like arrangement. Ensure that each wedge maintains a small gap from its neighbors, creating a central void and distinct separation between each body. The shape should result in a symmetric, modular cluster of six equally spaced, extruded rounded wedges.

Figure 7: Clarity and correctness: CADmium (light purple) accurately describes "six identical solid bodies, each shaped like a rounded wedge", whereas Text2CAD (light blue) provides a vague and insufficient description.

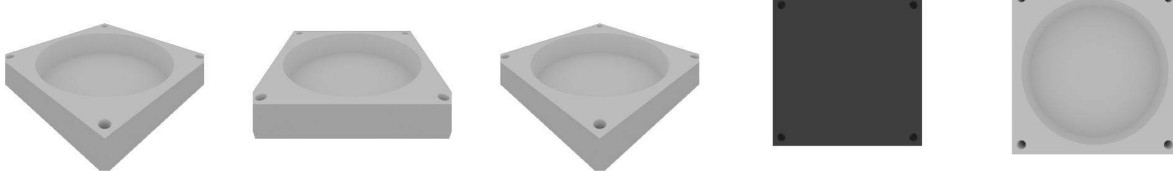

**Text2CAD Expert**
The first part of the CAD model is a three-dimensional, metallic, rectangular prism with a cylindrical hole in the center. To create this part, first set up a new coordinate system with Euler angles of (0.0, 0.0, 0.0) and a translation vector of (0.0, 0.0, 0.0).
Now, create a sketch on the X-Y plane of the coordinate system, consisting of six loops on a shared plane: loop 1, loop 2, loop 3, loop 4, loop 5 and loop 6. Loop 1 is a rectangular shape defined by four lines, loop 2 and loop 3 are circles with the same center at (0.0417, 0.0417) and (0.0417, 0.7083) respectively, loop 4 is a larger circle with center at (0.375, 0.375) and a radius of 0.3333, while loop 5 and loop 6 are circles similar to loop 2 and loop 3 but with centers located at (0.7083, 0.0417) and (0.7083, 0.7083) respectively.
Scale the 2D sketch by a factor of 0.75, then transform it into a 3D sketch using the euler angles and translation vector from the coordinate system. Extrude the 2D sketch to generate a 3D model with the following properties: extrusion depth towards normal is 0.1667, opposite the normal is 0.0, and the sketch scale is 0.75. This operation creates a new solid body.
The part has the following dimensions: length is **0.7499999999999997**, width is **0.7499999999999997**, and height is **0.16666666666666663** (including the extrusion depth).
For the second part of the CAD model, create a new coordinate system with Euler angles of (0.0, 0.0, 0.0) and a translation vector of (0.0417, 0.0417, 0.0).
Create a sketch on a plane defined by the coordinate system, consisting of a single loop: loop 1. This loop is defined by a circle with center at (0.3333, 0.3333) and radius of 0.3333.
Scale the 2D sketch by a factor of 0.6667, then transform it into a 3D sketch using the euler angles and translation vector from the second coordinate system. Extrude the 2D sketch to generate a 3D model with the following properties: extrusion depth towards normal is 0.0333, opposite the normal is 0.0, and the sketch scale is 0.6667. This operation joins with the previous body.
The part has the following dimensions: length is **0.6666666666666665**, width is **0.6666666666666665**, and height is **0.033333333333333326** (including the extrusion depth).
The final shape is a three-dimensional, rectangular prism with a curved, hollowed-out center. The dimensions of the prism are approximately 20 units long, 10 units wide, and 5 units high (including the height of the cylindrical hole).

**CADmium**
Begin by creating a solid cylinder with a diameter of 0.4167 and a height of 0.3333. On the top face of the cylinder, sketch a circle centered at (0.2083, 0.2083) with a radius of 0.2083. Inside this circle, define a closed loop comprised of two horizontal lines and an arc: draw the first line from (0.1383, 0.1176) to (0.1994, 0.1176), the second line from (0.1994, 0.1176) to (0.2783, 0.1176), and connect the end points of these lines back to the starting point with an arc that has its midpoint at (0.2083, 0.3229), forming a segment that cuts across the base of the circle. Extrude this internal profile fully through the cylinder to create a cylindrical hole with a flat chord across its lower portion as defined by the lines and arc, resulting in a cylinder with a partial circular cutout running through its center.

Figure 8: Numerical representation: CADmium (light purple) uses natural numerical precision, contrasting with Text2CAD's (light blue) use of overly precise, unnatural decimals.

# B    Metrics on a Common Evaluable Subset

As detailed in Section 4, the calculation of various evaluation metrics is contingent upon the generated CAD models meeting specific validity and structural prerequisites. The number of samples satisfying these diverse requirements can vary across different models or experimental setups. The results presented in the main paper maximize data utilization by reporting metrics on all evaluable samples for each individual condition, which can lead to comparisons across slightly different sets of underlying CAD models.

To provide a more direct and controlled comparison, this appendix presents supplementary tables corresponding to the key experiments in the main paper. In these tables, all metrics are re-calculated based exclusively on the intersection of CAD models that were successfully generated and met all necessary criteria (including validity and any specific prerequisites like watertightness) across all experimental conditions being compared within that specific analysis. This common subset evaluation ensures that different models or configurations are assessed on an identical cohort of generated objects, offering clearer insights into relative performance by minimizing confounding effects from varying sample populations.

Table 5: Performance comparison on the human-annotated *CADPrompt* dataset (common subset evaluation). Metrics are computed on the intersection of samples for which both models produced valid CAD sequences. Reported values include 95% confidence intervals in brackets. Boldface indicates the better-performing model whenever the difference is statistically significant at $p < 0.05$ (Wilcoxon signed-rank test for continuous metrics, McNemar's test for the binary watertightness metric).

| Model
Trained on | Text2CAD
Text2CAD | Qwen-2.5 Coder 14B
CADmium |
|---|---|---|
| IR (%) ↓ | **1.57** | 6.28 |
| Num Valid | 176 | 176 |
| Line F1 ↑ | 0.67 [0.63, 0.72] | 0.69 [0.64, 0.74] |
| Arc F1 ↑ | 0.00 [0.00, 0.00] | **0.23** [0.11, 0.36] |
| Circle F1 ↑ | 0.33 [0.23, 0.42] | **0.61** [0.51, 0.71] |
| Extrusion F1 ↑ | 0.84 [0.82, 0.87] | **0.89** [0.87, 0.92] |
| CD mean ↓ | 224.38 [187.33, 263.47] | 203.04 [173.25, 235.10] |
| CD median ↓ | 129.47 [87.59, 170.25] | 112.84 [78.86, 177.11] |
| SIR ↓ | **0.03** [0.01, 0.05] | 0.10 [0.07, 0.15] |
| DangEL ↓ | **0.21** [0.03, 0.47] | 2.99 [1.67, 4.64] |
| SegE ↓ | **0.10** [0.03, 0.18] | 0.78 [0.48, 1.14] |
| FluxEE ($\times 10^2$) ↓ | **0.05** [0.00, 0.12] | 2.28 [1.04, 3.78] |
| Watertightness (%) ↑ | **95.74** [92.55, 98.40] | 83.80 [78.21, 88.83] |
| Num Watertight | 128 | 128 |
| EECM ↑ | 0.66 [0.55, 0.71] | [0.59, 0.75] |
| DMCD ($\times 10^3$) ↓ | 14.88 [12.74, 17.12] | **8.34** [6.54, 10.32] |
| SD mean ($\times 10^2$) ↓ | 15.69 [13.37, 18.27] | **7.22** [5.09, 9.60] |
| SD median ($\times 10^2$) ↓ | 12.26 [9.26, 15.14] | **1.51** [1.06, 2.15] |

Table 6: Impact of Qwen Coder model scale (1.5B, 3B, 7B, 14B parameters) on the **CADmium test set** (common subset evaluation). For each dataset, metrics are computed on the intersection of samples for which all three model sizes produced valid CAD sequences. All models were trained on CADmium annotations. Reported values include 95% confidence intervals in brackets. Statistical significance is assessed with statistical tests on adjacent sizes (1.5B→3B, 3B→7B, 7B→14B). Bold indicates a size that is significantly better than the next smaller size at $p < 0.05$. Details of the statistical testing procedure are provided in Section 4.

| Metric | 1.5B | 3B | 7B | 14B |
|---|---|---|---|---|
| IR (%) ↓ | 11.86 | 8.51 | 5.93 | 3.33 |
| Num Valid | 6500 | 6500 | 6500 | 6500 |
| Line F1 ↑ | 0.93 [0.92, 0.93] | **0.94** [0.93, 0.94] | **0.94** [0.94, 0.95] | **0.95** [0.95, 0.95] |
| Arc F1 ↑ | 0.78 [0.76, 0.81] | **0.80** [0.77, 0.82] | **0.82** [0.79, 0.84] | **0.86** [0.84, 0.88] |
| Circle F1 ↑ | 0.90 [0.89, 0.91] | **0.93** [0.92, 0.94] | 0.93 [0.92, 0.94] | **0.95** [0.94, 0.95] |
| Extrusion F1 ↑ | 0.99 [0.98, 0.99] | 0.99 [0.99, 0.99] | **0.99** [0.99, 0.99] | 0.99 [0.99, 0.99] |
| CD mean ↓ | 64.74 [60.84, 68.67] | **39.48** [36.03, 42.97] | 54.23 [50.43, 58.22] | **37.11** [34.13, 40.33] |
| CD median ↓ | 0.26 [0.25, 0.27] | 0.21 [0.21, 0.22] | 0.23 [0.21, 0.24] | **0.20** [0.20, 0.21] |
| SIR ↓ | 0.04 [0.03, 0.04] | **0.03** [0.03, 0.03] | 0.03 [0.02, 0.03] | 0.02 [0.02, 0.03] |
| DangEL ↓ | 0.94 [0.80, 1.09] | **0.76** [0.65, 0.87] | **0.65** [0.56, 0.76] | 0.57 [0.48, 0.67] |
| SegE ↓ | 0.33 [0.29, 0.38] | **0.30** [0.26, 0.34] | 0.26 [0.23, 0.29] | **0.24** [0.21, 0.28] |
| FluxEE ($\times 10^2$) ↓ | 0.30 [0.23, 0.38] | **0.30** [0.22, 0.38] | 0.24 [0.17, 0.32] | **0.25** [0.18, 0.32] |
| Watertightness (%) ↑ | 90.38 [89.69, 91.06] | **90.31** [89.62, 90.99] | **90.54** [89.87, 91.19] | **90.78** [90.13, 91.44] |
| Num Watertight | 6205 | 6408 | 6575 | 6779 |
| EECM ↑ | 0.90 [0.89, 0.91] | **0.91** [0.91, 0.92] | 0.91 [0.90, 0.92] | **0.92** [0.92, 0.93] |
| DMCD ($\times 10^3$) ↓ | 2.42 [2.25, 2.60] | **2.18** [2.00, 2.37] | 2.08 [1.91, 2.26] | **1.95** [1.78, 2.14] |
| SD mean ($\times 10^2$) ↓ | 2.51 [2.31, 2.72] | **1.83** [1.67, 1.99] | 1.82 [1.66, 1.99] | **1.60** [1.44, 1.76] |
| SD median ($\times 10^2$) ↓ | 0.00 [0.00, 0.00] | 0.00 [0.00, 0.00] | 0.00 [0.00, 0.00] | 0.00 [0.00, 0.00] |

Table 7: Impact of Qwen Coder model scale (1.5B, 3B, 7B, 14B parameters) on the **Fusion360 test set** (common subset evaluation). For each dataset, metrics are computed on the intersection of samples for which all three model sizes produced valid CAD sequences. All models were trained on CADmium annotations. Reported values include 95% confidence intervals in brackets. Statistical significance is assessed with statistical tests on adjacent sizes (1.5B→3B, 3B→7B, 7B→14B). Bold indicates a size that is significantly better than the next smaller size at $p < 0.05$. Details of the statistical testing procedure are provided in Section 4.

| Metric | 1.5B | 3B | 7B | 14B |
|---|---|---|---|---|
| IR (%) ↓ | 22.62 | 23.38 | 16.79 | 12.58 |
| Num Valid | 5304 | 5304 | 5304 | 5304 |
| Line F1 ↑ | 0.86 [0.85, 0.87] | **0.88** [0.87, 0.89] | **0.90** [0.89, 0.91] | **0.91** [0.90, 0.92] |
| Arc F1 ↑ | 0.74 [0.71, 0.76] | **0.75** [0.73, 0.77] | **0.78** [0.76, 0.80] | **0.81** [0.79, 0.83] |
| Circle F1 ↑ | 0.90 [0.89, 0.91] | **0.92** [0.92, 0.93] | 0.92 [0.92, 0.93] | **0.94** [0.93, 0.94] |
| Extrusion F1 ↑ | 0.98 [0.98, 0.99] | 0.98 [0.98, 0.98] | **0.99** [0.99, 0.99] | 0.99 [0.99, 0.99] |
| CD mean ↓ | 227.21 [218.41, 236.09] | **188.94** [181.02, 197.01] | 206.18 [197.73, 214.94] | 211.93 [202.79, 221.16] |
| CD median ↓ | 103.82 [97.32, 110.25] | 72.37 [64.80, 80.78] | 85.95 [79.99, 91.91] | 73.71 [63.45, 82.08] |
| SIR ↓ | 0.04 [0.03, 0.04] | 0.04 [0.03, 0.04] | 0.05 [0.04, 0.05] | **0.03** [0.03, 0.03] |
| DangEL ↓ | 1.03 [0.90, 1.15] | **0.86** [0.73, 0.98] | 1.21 [1.06, 1.37] | **0.73** [0.61, 0.85] |
| SegE ↓ | 0.37 [0.34, 0.41] | **0.34** [0.30, 0.38] | 0.38 [0.34, 0.42] | **0.28** [0.25, 0.32] |
| FluxEE ($\times 10^2$) ↓ | 0.20 [0.15, 0.27] | **0.20** [0.14, 0.27] | 0.18 [0.12, 0.25] | 0.18 [0.12, 0.23] |
| Watertightness (%) ↑ | 86.62 [85.78, 87.46] | 86.50 [85.67, 87.35] | **85.21** [84.38, 86.04] | **87.35** [86.56, 88.10] |
| Num Watertight | 5241 | 5192 | 5504 | 5941 |
| EECM ↑ | 0.82 [0.81, 0.83] | **0.84** [0.83, 0.86] | 0.84 [0.83, 0.85] | **0.87** [0.86, 0.88] |
| DMCD ($\times 10^3$) ↓ | 4.04 [3.78, 4.30] | **3.36** [3.12, 3.61] | 3.56 [3.31, 3.81] | **3.26** [3.02, 3.52] |
| SD mean ($\times 10^2$) ↓ | 3.55 [3.29, 3.82] | **2.53** [2.33, 2.75] | 2.91 [2.68, 3.15] | **2.51** [2.29, 2.73] |
| SD median ($\times 10^2$) ↓ | 0.00 [0.00, 0.00] | 0.00 [0.00, 0.00] | 0.00 [0.00, 0.00] | 0.00 [0.00, 0.00] |

Table 8: Impact of annotation quality using the Text2CAD architecture (common subset evaluation). Models were trained on either CADmium or Text2CAD annotations. For each of the four training/evaluation set pairings, metrics are computed on the intersection of samples for which both compared conditions produced valid CAD sequences. Reported values include 95% confidence intervals in brackets. Boldface indicates statistically significant differences ($p < 0.05$); details of the statistical testing procedure are provided in Section 4.

| Evaluated on | CADmium | | Text2CAD | |
|---|---|---|---|---|
| Trained on | CADmium | Text2CAD | CADmium | Text2CAD |
| IR (%) ↓ | **3.07** | 4.44 | 4.95 | **4.34** |
| Num Valid | 7277 | 7277 | 7469 | 7469 |
| Line F1 ↑ | **0.89** [0.89, 0.90] | 0.74 [0.73, 0.75] | 0.64 [0.64, 0.65] | **0.83** [0.82, 0.83] |
| Arc F1 ↑ | **0.60** [0.58, 0.62] | 0.22 [0.20, 0.24] | 0.25 [0.23, 0.27] | **0.38** [0.36, 0.41] |
| Circle F1 ↑ | **0.89** [0.88, 0.90] | 0.58 [0.57, 0.60] | 0.47 [0.46, 0.49] | **0.76** [0.75, 0.77] |
| Extrusion F1 ↑ | **0.98** [0.98, 0.98] | 0.90 [0.89, 0.90] | 0.81 [0.80, 0.81] | **0.94** [0.93, 0.94] |
| CD mean ↓ | **50.44** [46.99, 53.92] | 144.85 [140.06, 149.69] | 107.47 [103.45, 111.48] | **28.52** [26.58, 30.47] |
| CD median ↓ | **0.31** [0.29, 0.32] | 50.78 [46.96, 55.46] | 33.38 [30.78, 35.88] | **0.37** [0.34, 0.40] |
| SIR ↓ | 0.04 [0.04, 0.05] | 0.04 [0.04, 0.05] | 0.08 [0.07, 0.08] | 0.04 [0.04, 0.05] |
| DangEL ↓ | 1.36 [1.22, 1.51] | **1.11** [0.97, 1.26] | 2.38 [2.17, 2.60] | **1.06** [0.94, 1.19] |
| SegE ↓ | 0.63 [0.53, 0.78] | 0.50 [0.44, 0.57] | 1.06 [0.97, 1.16] | **0.48** [0.44, 0.53] |
| FluxEE ($\times 10^2$) ↓ | 0.39 [0.31, 0.47] | 0.57 [0.45, 0.71] | 0.93 [0.80, 1.06] | **0.42** [0.33, 0.53] |
| Watertightness (%) ↑ | 87.63 [86.88, 88.36] | **91.90** [91.29, 92.50] | 82.48 [81.62, 83.34] | **90.52** [89.85, 91.17] |
| Num Watertight | 6033 | 6033 | 5673 | 5673 |
| EECM ↑ | **0.87** [0.86, 0.88] | 0.77 [0.78, 0.80] | 0.66 [0.66, 0.68] | **0.82** [0.83, 0.85] |
| DMCD ($\times 10^3$) ↓ | **3.37** [3.16, 3.59] | 7.64 [7.32, 7.95] | 9.19 [8.89, 9.50] | **5.09** [4.81, 5.36] |
| SD mean ($\times 10^2$) ↓ | **2.50** [2.34, 2.68] | 8.31 [8.01, 8.62] | 11.31 [10.91, 11.71] | **4.68** [4.42, 4.95] |
| SD median ($\times 10^2$) ↓ | **0.00** [0.00, 0.00] | 3.20 [2.87, 3.48] | 5.07 [4.72, 5.48] | **0.02** [0.00, 0.05] |

## C  Prompt Details

This appendix details the various prompts utilized in our research pipeline.

The generation of our CADmium annotations was performed using GPT-4.1. The system message provided to GPT-4.1, which guided it to produce the descriptions, is displayed in Figure 9. The user message is composed of the minimal JSON representation of the CAD model, 10 multi-view images rendered with Blender, and the instructions.

For the text-to-CAD generation task, our fine-tuned Qwen Coder models take a natural language description as input. The specific format used to instruct the model to generate the corresponding JSON-based CAD sequence is shown in Figure 10.

To evaluate the quality of our CADmium annotations against CADLLM and Text2CAD expert annotations, we employed an LLM-as-a-judge setup using both Gemma-3 12B (Team et al., 2025) and Mistral 3.2 Small (Mistral, 2025), as described in Section 3.1 and with results presented in Figure 2b. In this evaluation, 10,000 paired annotations (one from CADmium, one for CADLLM, one from Text2CAD expert for the same CAD model) were assessed. To mitigate positional bias, the annotations were randomly assigned to the labels "1.", "2.", and "3." within the user message presented to Gemma-3 12B. For optimizing the classification process and minimizing output variability, the model was explicitly instructed to respond with only a single digit (0, 1, 2, or 3), allowing the judgment to be captured efficiently. The specific prompts used to instruct LLMs for each of the four evaluation criteria are detailed as follows:

- The prompt for assessing **human-likeness** is shown in Figure 11.

- The prompt for assessing **clarity and readability** is shown in Figure 14.

- The prompt for assessing **visual faithfulness** (given image renders of the object) is shown in Figure 13.

- The prompt for assessing **completeness** against the ground truth minimal JSON description is shown in Figure 12.

**User Message**

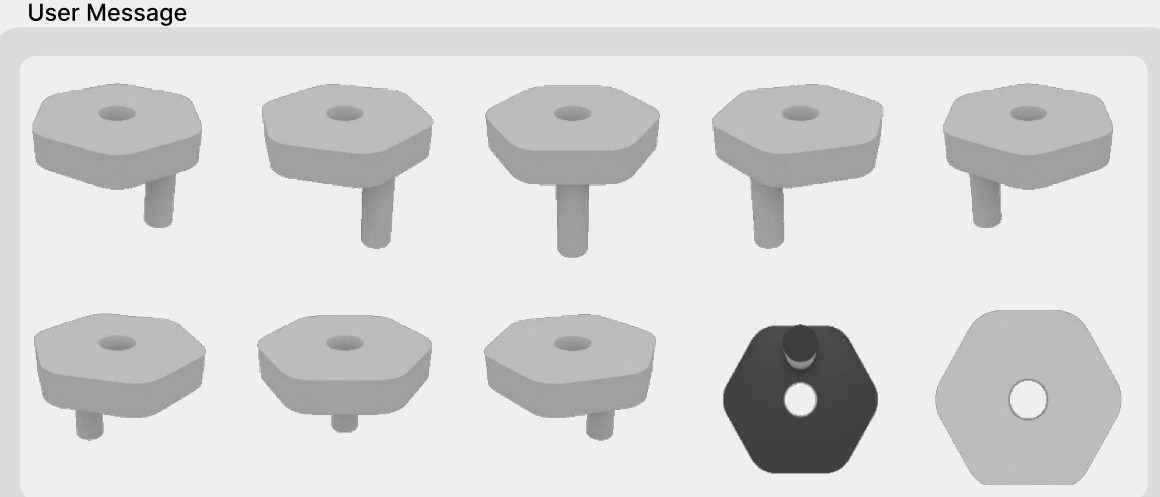

You are an expert mechanical engineer tasked with creating clear, precise instructions for a text-to-CAD generator.
I have a set of 9 multi-view images displaying a 3D model, as well as a JSON file describing the exact CAD operations
used to construct the object.

This is the json file:
```json
{"parts": {
    "part_1": {
      "coordinate_system": { [ ... ] },
      "sketch": { [ ... ] },
      "extrusion": { [ ... ] }
    "part_2": { [ ... ] }
}}}
```

Create a single, comprehensive text description of this 3D object that:
- Describes all geometrical features accurately based on the operations and dimensions
- Uses natural language as if a human designer were explaining how to model this object
- Is written in second-person as instructions for a text-to-CAD system
- Includes all critical dimensions and geometric relationships (note that you don't need to specify the unit of measurement
for lengths)
- Avoids redundancy while ensuring completeness
- Focuses on the design intent and functional geometry
- Answer only with the description. No introductory phrases, titles, commentary, summaries or conclusions

Your description should be concise but complete, capturing every important geometric feature without unnecessary
repetition.

Figure 9: Prompt structure for CAD model annotation. The user message contains the JSON description and the 10 multi-view renders of the 3D models, and instruct GPT-4.1 to generate a natural language description.

**System Message**

```
Generate CAD model JSON EXACTLY matching this schema:

{
  "parts": {
    "part_1": {  // Always use sequential part_1, part_2... even if some are null
      "coordinate_system": {
        "Euler Angles": [0.0, 0.0, 0.0],  // XYZ rotation angles in degrees
        "Translation Vector": [0.0, 0.0, 0.0]  // X,Y,Z position offsets
      },
      "description": {
        "height": 0.0,  // Total vertical dimension
        "length": 0.0,  // Total horizontal dimension
        "name": "",    // (optional) Component identifier
        "shape": "",   // (optional) Basic geometric classification
        "width": 0.0   // Total depth dimension
      },
      "extrusion": {
        "extrude_depth_opposite_normal": 0.0,  // Negative direction extrusion
        "extrude_depth_towards_normal": 0.0,   // Positive direction extrusion
        "operation": "NewBodyFeatureOperation", // One of: NewBodyFeatureOperation, JoinFeatureOperation,
CutFeatureOperation, IntersectFeatureOperation
        "sketch_scale": 0.0  // Scaling factor for sketch geometry
      },
      "sketch": {
        "face_1": {  // Use sequential face_1, face_2... (null if unused)
          "loop_1": {  // Use sequential loop_1, loop_2... (null if unused)
            "circle_1": {  // Use sequential circle_1, circle_2...
              "Center": [0.0, 0.0],  // X,Y coordinates
              "Radius": 0.0
            },
            "arc_1": {  // Use sequential arc_1, arc_2...
              "Start Point": [0.0, 0.0],
              "End Point": [0.0, 0.0],
              "Mid Point": [0.0, 0.0]
            },
            "line_1": {  // Use sequential line_1, line_2...
              "Start Point": [0.0, 0.0],
              "End Point": [0.0, 0.0]
            }
            // ... (other geometric elements as null/none)
          }
          // ... (other loops as null/none)
        }
        // ... (other faces as null/none)
      }
    },
    "part_2": null,  // Maintain sequential numbering even for null parts
    // ... (additional parts)
  }
}

STRICT RULES:
- OUTPUT ONLY RAW JSON (no formatting/text/comments/explanations)
- NEVER COPY INSTRUCTIONAL TEXT FROM JSON SCHEMA EXAMPLES
- ALL numbers as floats (0.0 not 0)
- ALLOWED OPERATIONS: NewBodyFeatureOperation/JoinFeatureOperation/CutFeatureOperation/
IntersectFeatureOperation
- GEOMETRY REQUIREMENTS (these are the only available primitives):
  • Circles: Center[X,Y] + Radius
  • Arcs: Start[X,Y] + End[X,Y] + Mid[X,Y]
  • Lines: Start[X,Y] + End[X,Y]
- ENFORCE part_1, part_2... sequence (include nulls)
- NO NEW FIELDS
```

**User Message**

Begin by creating a solid cylinder with a diameter of 0.4167 and a height of 0.3333. On the top face of the cylinder, sketch a circle centered at (0.2083, 0.2083) with a radius of 0.2083. Inside this circle, define a closed loop comprised of two horizontal lines and an arc: draw the first line from (0.1383, 0.1176) to (0.1994, 0.1176), the second line from (0.1994, 0.1176) to (0.2783, 0.1176), and connect the end points of these lines back to the starting point with an arc that has its midpoint at (0.2083, 0.3229), forming a segment that cuts across the base of the circle. Extrude this internal profile fully through the cylinder to create a cylindrical hole with a flat chord across its lower portion as defined by the lines and arc, resulting in a cylinder with a partial circular cutout running through its center.

Figure 10: Prompt structure for CAD model generation. The system message specifies the JSON schema and constraints, while the user message contains the natural language description of the target CAD model.

**System Message**

You are an expert linguistic analyst. Your task is to determine which of three provided text annotations describing a 3D CAD model is more likely to have been written by a human.
You will be presented with three annotations, labeled "1.", "2.", and "3.".
Respond with:
- "1" if Annotation 1 is more likely human-generated.
- "2" if Annotation 2 is more likely human-generated.
- "3" if Annotation 3 is more likely human-generated.
- "0" if all are equally likely human-generated, all seem AI-generated, or you cannot confidently determine a difference.
Your entire response must be a single digit (0, 1, 2, or 3). Do not provide any other text, explanation, or punctuation.

**User Message**

Here are two annotations for a 3D CAD model:
1. "{annotation_1}"
2. "{annotation_2}"
3. "{annotation_3}"
Which annotation is more likely human-generated? If you cannot confidently determine, or if they seem equally likely human or AI, indicate a tie. Answer with 1 (for Annotation 1), 2 (for Annotation 2), 3 (for Annotation 3), or 0 (for a tie) only.

Figure 11: Prompt for the LLM-as-a-judge evaluation, tasking the judge to assess and compare the human-likeness of two provided CAD model descriptions (one from CADmium, one from Text2CAD expert).

**System Message**

You are an expert 3D CAD model analyst with strong natural language understanding capabilities.
Your task is to evaluate three text annotations against a detailed JSON description of a 3D CAD object's construction history and features.
The JSON describes the object in terms of its constituent parts, sketches (including geometric primitives like lines and arcs), extrusion operations, coordinate systems, and descriptive properties (like name, shape, and dimensions).
You must determine which of the two text annotations more completely, accurately, and exhaustively represents these core features as detailed in the provided JSON.
You will be given the JSON data and three annotations labeled "1.", "2.", and "3.".
Respond with:
- "1" if Annotation 1 is more comprehensive and accurate in representing the JSON's core features.
- "2" if Annotation 2 is more comprehensive and accurate in representing the JSON's core features.
- "3" if Annotation 3 is more comprehensive and accurate in representing the JSON's core features.
- "0" if all annotations are approximately equal in their representation of the JSON's core features, or if all are equally lacking.
Your entire response must be a single digit (0, 1, 2, or 3). Do not provide any other text, explanation, or punctuation.

**User Message**

The following is a JSON description detailing the construction and features of a 3D CAD object:
```json
{"parts": {
    "part_1": {
      "coordinate_system": { [ ... ] },
      "sketch": { [ ... ] },
      "extrusion": { [ ... ] }
    "part_2": { [ ... ] }
}}}
```

Now, consider these three text annotations for the same 3D CAD object:
"{annotation_1}"
"{annotation_2}"
"{annotation_3}"
Based on the detailed information in the JSON description provided above, which annotation (1, 2, or 3) most exhaustively and accurately represents the core features of the 3D CAD object (such as its parts, their geometric makeup, construction operations, and key descriptive properties)? If all are substantially equal in this regard or equally incomplete, indicate a tie. Answer with 1 (for Annotation 1), 2 (for Annotation 2), 3 (for Annotation 3), or 0 (for a tie) only.

Figure 12: Prompt for the LLM-as-a-judge evaluation, tasking the judge to assess and compare the completeness of two provided CAD model descriptions against the ground truth minimal JSON representation of the corresponding 3D object.

Figure 13: Prompt for the LLM-as-a-judge evaluation, tasking the judge to assess and compare the visual faithfulness of two provided CAD model descriptions against multi-view image renders of the corresponding 3D object.

### System Message

You are an expert visual-textual analyst. Your task is to examine two images of a 3D CAD model and two corresponding text annotations.
You must determine which of the three annotations most accurately and faithfully describes the visual content presented in the images.
You will be provided with three annotations, labeled "1.", "2.", and "3.". The images will be provided alongside this text.
Respond with:
- "1" if Annotation 1 most accurately describes the images.
- "2" if Annotation 2 most accurately describes the images.
- "3" if Annotation 3 most accurately describes the images.
- "0" if all annotations describe the images with equal accuracy, or if neither adequately describes the images, or you cannot confidently determine a difference.
Your entire response must be a single digit (0, 1, 2, or 3). Do not provide any other text, explanation, or punctuation.

### User Message

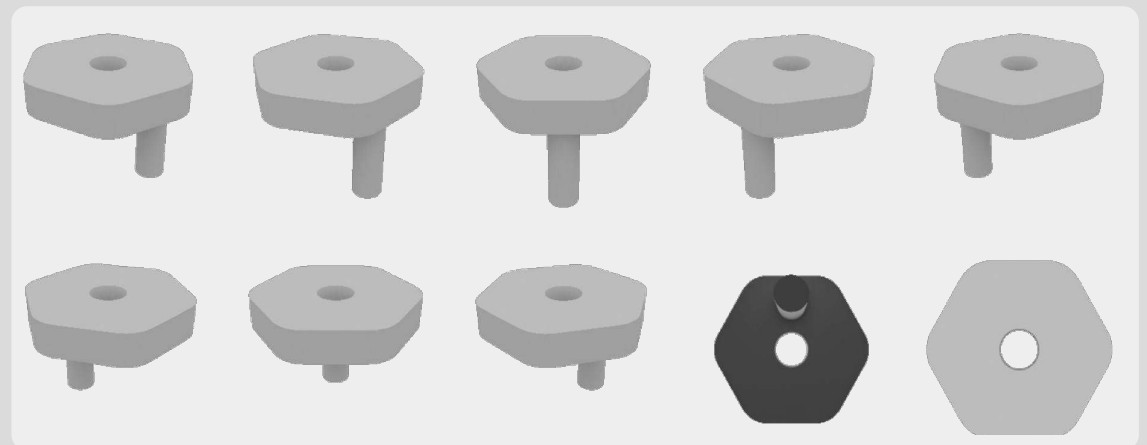

You are viewing two images of a 3D CAD model.
Below are two text annotations:
1. "{annotation_1}"
2. "{annotation_2}"
3. "{annotation_3}"
Considering the visual details in the provided images, which annotation (1 or 2) most accurately and faithfully describes the 3D model shown? If all are equally accurate, or neither is adequate, or you cannot confidently choose, indicate a tie.
Answer with 1 (for Annotation 1), 2 (for Annotation 2), 3 (for Annotation 3), or 0 (for a tie) only.

**System Message**

You are an expert linguistic evaluator. Your task is to analyze two text annotations provided for a 3D CAD model and determine which one exhibits superior clarity, conciseness, and overall readability for a general audience.
You will be presented with three annotations, labeled "1." and "2.", and "3.".
Consider factors such as clear language, straightforward sentence structure, and ease of understanding.
Respond with:
- "1" if Annotation 1 is clearer and more readable.
- "2" if Annotation 2 is clearer and more readable.
- "3" if Annotation 3 is clearer and more readable.
- "0" if all annotations are approximately equal in clarity and readability, or if all are equally unclear.
Your entire response must be a single digit (0, 1, 2, or 3). Do not provide any other text, explanation, or punctuation.

**User Message**

Below are three text annotations for a 3D CAD model. Please evaluate them based on their clarity, conciseness, and overall readability.
1. "{annotation_1}"
2. "{annotation_2}"
3. "{annotation_3}"
Which annotation is clearer, more concise, and easier for a general audience to understand? If they are substantially equal in these aspects, or equally unclear, indicate a tie. Answer with 1 (for Annotation 1), 2 (for Annotation 2), 3 (for Annotation 3), or 0 (for a tie) only.

Figure 14: Prompt for the LLM-as-a-judge evaluation, tasking the judge to assess and compare the clarity and readability of two provided CAD model descriptions (one from CADmium, one from Text2CAD expert).

# D  Qualitative Analysis of New Metrics

The purpose of this section is to show examples of cases where the new metrics (EECM, SD, and DMCD) had poor values for the test dataset reconstructions by Qwen-7B model trained on the CADmium dataset.

**Exact Euler Characteristic Match (EECM)**

### Ground Truth, UID: 00262126

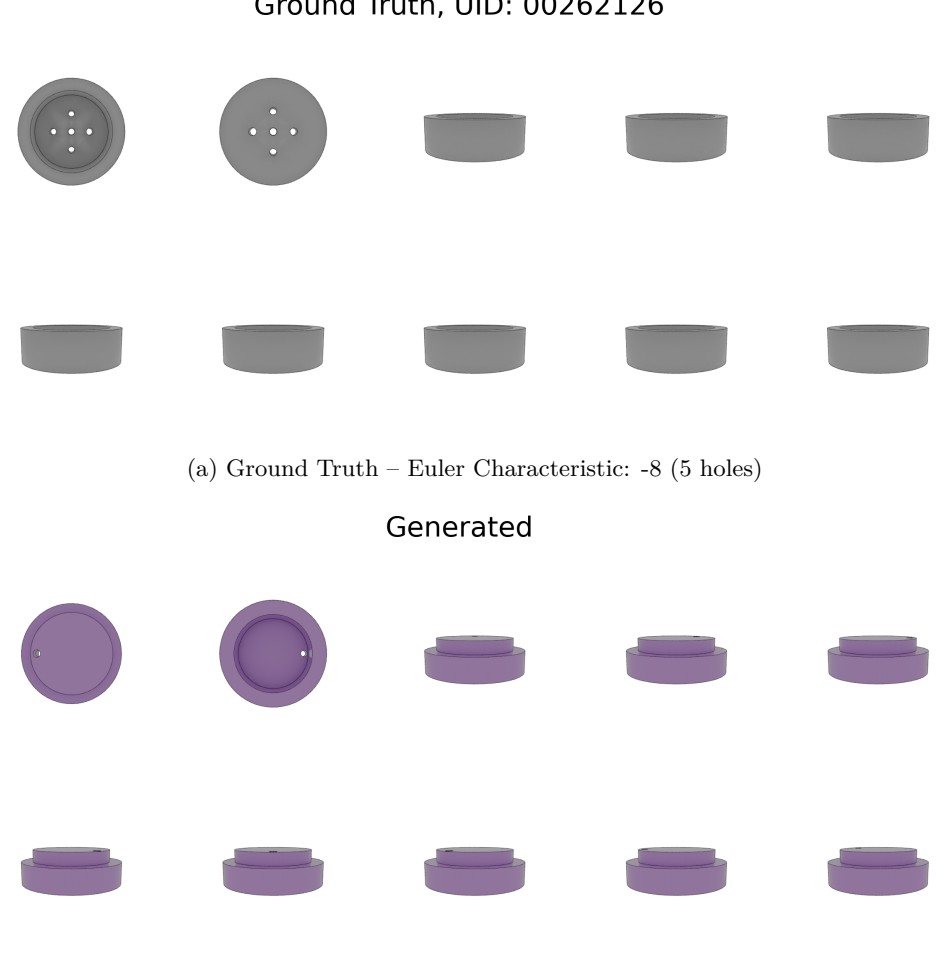

(a) Ground Truth – Euler Characteristic: -8 (5 holes)

### Generated

(c) Generated – Euler Characteristic: 0 (1 hole)

Figure 15: In this example, the Euler Score mismatch translates to the discrepancy in the number of holes between the ground truth (5 holes) and the generated reconstruction (1 hole).

Ground Truth, UID: 00260276

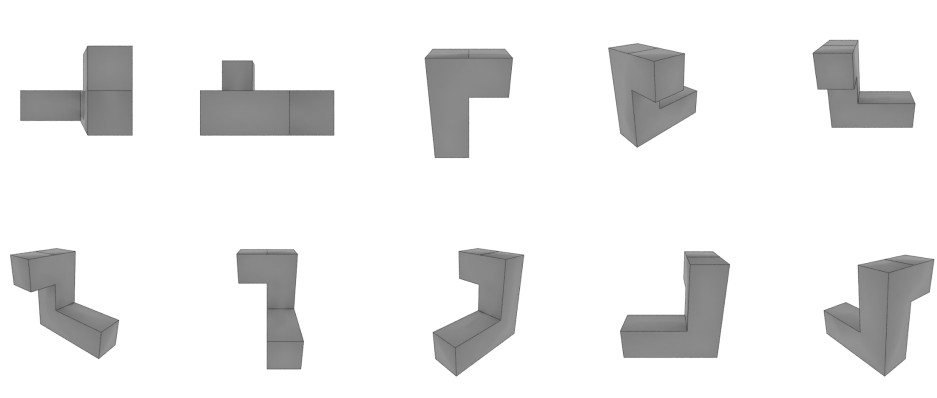

(a) Ground Truth – Euler Characteristic: 2 (0 holes, 1 connected component)

Generated

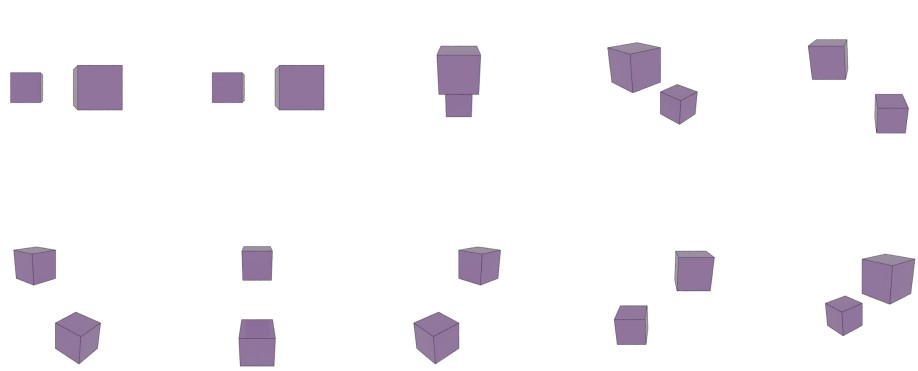

(c) Generated – Euler Characteristic: 4 (0 holes, 2 connected components)

Figure 16: In this example, the Euler Score mismatch translates to the discrepancy in the number of connected components, as there are two components in the generated reconstruction.

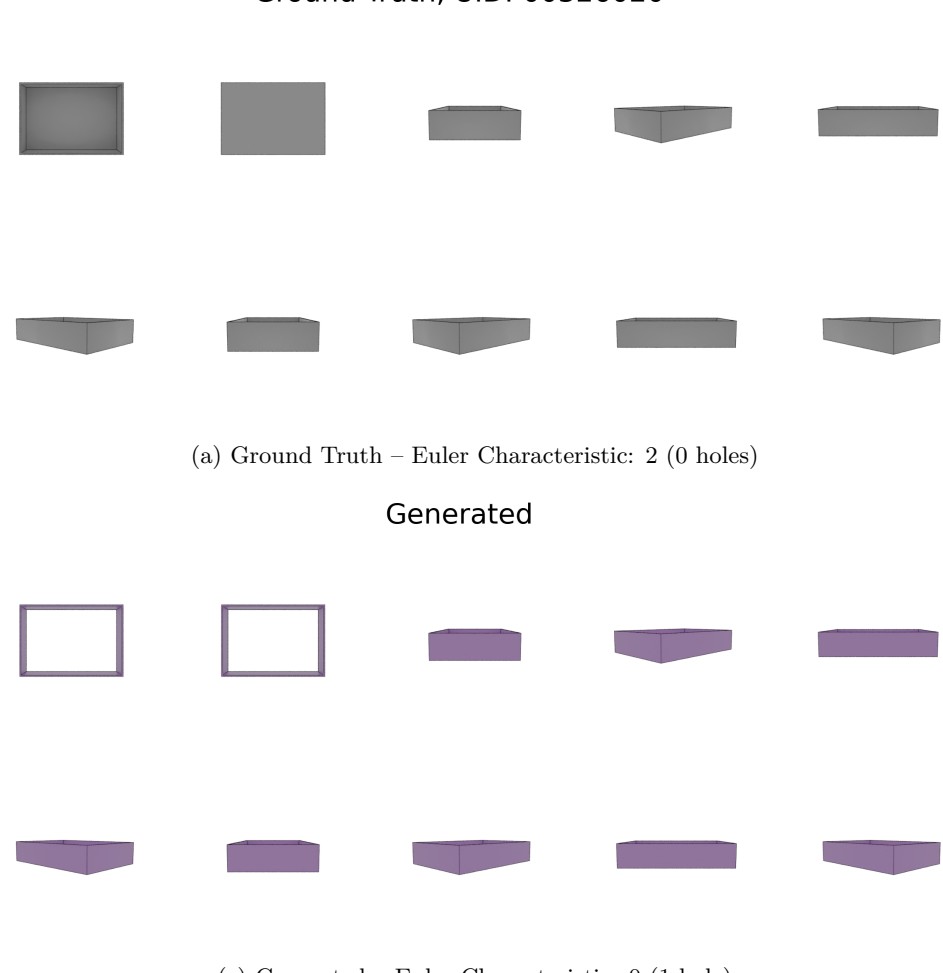

(a) Ground Truth – Euler Characteristic: 2 (0 holes)

(c) Generated – Euler Characteristic: 0 (1 hole)

Figure 17: In this example, the Euler Score mismatch translates to the discrepancy in the number of holes between the ground truth (0 holes) and the generated reconstruction (1 hole).

**Sphericity Discrepancy (SD)**

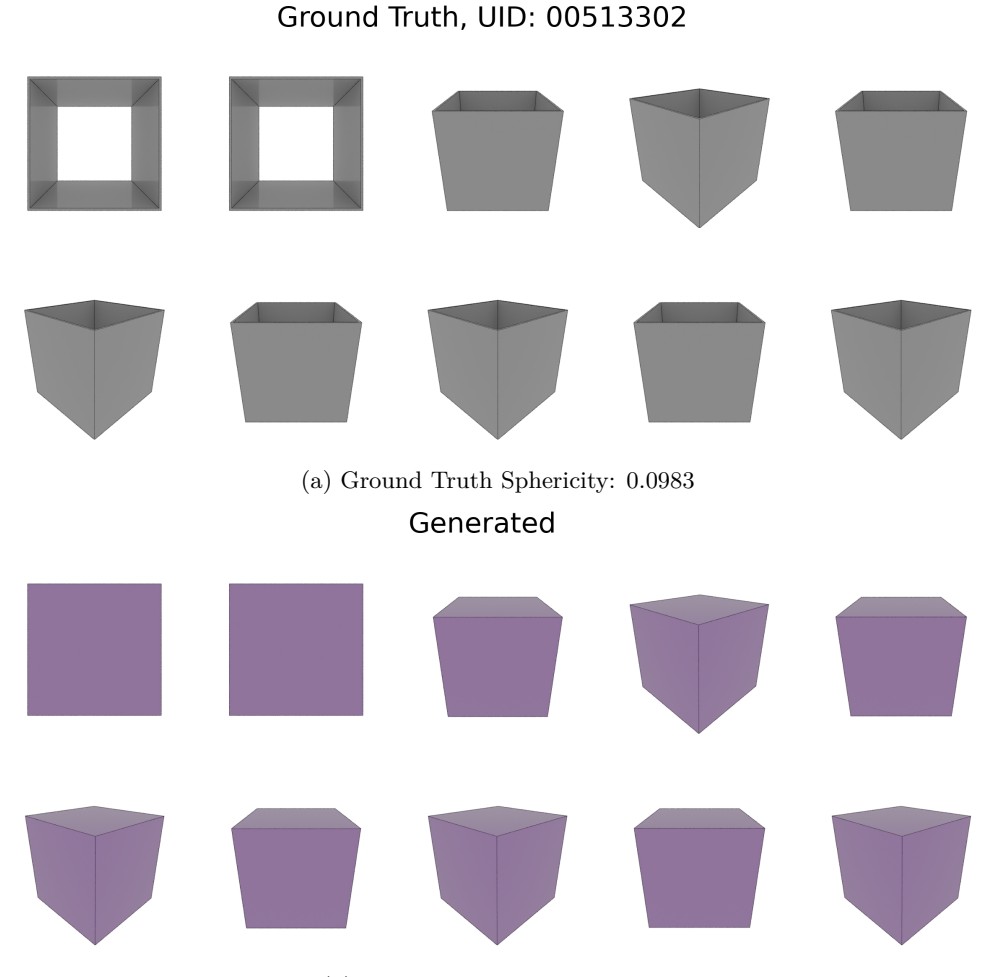

(a) Ground Truth Sphericity: 0.0983

(b) Generated Sphericity: 0.8059

Figure 18: **SD**: 0.7076; The discrepancy in the sphericity indicates the difference in the overall shape. Higher sphericity indicates closer resemblance to a sphere.

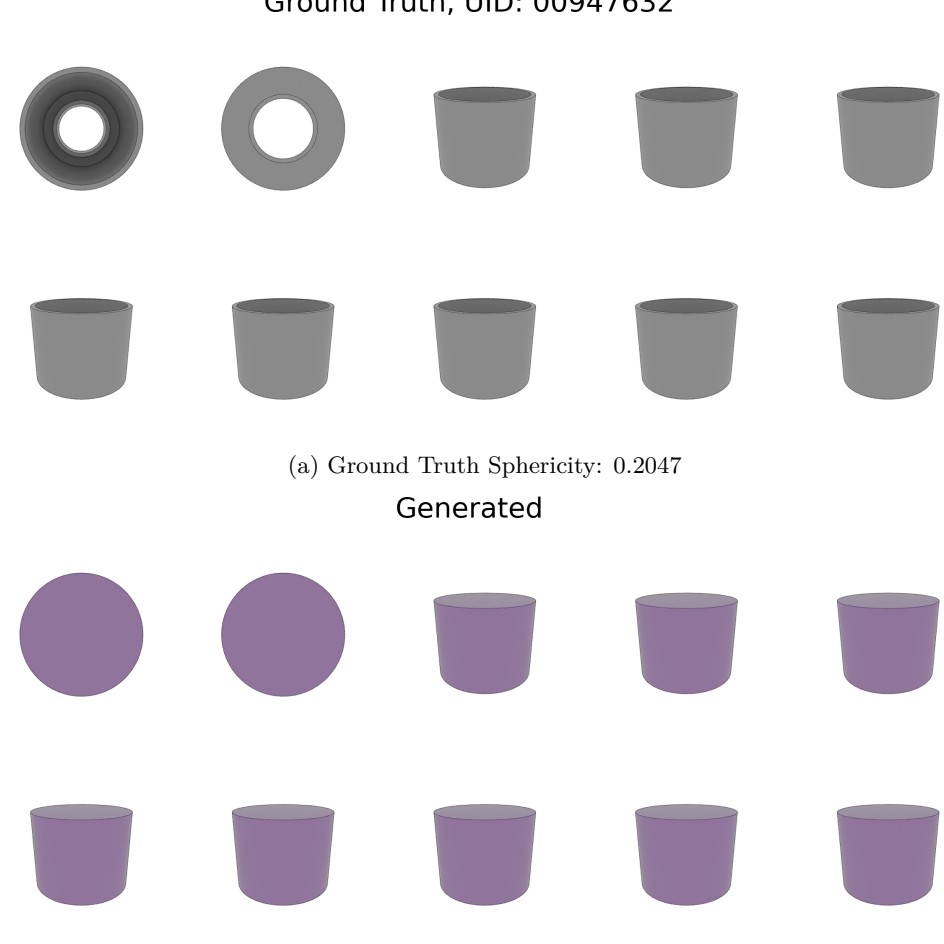

(a) Ground Truth Sphericity: 0.2047

(b) Generated Sphericity: 0.8667

Figure 19: **SD**: 0.6620; The discrepancy in the sphericity indicates the difference in the overall shape. Higher sphericity indicates closer resemblance to a sphere.

Ground Truth, UID: 00963721

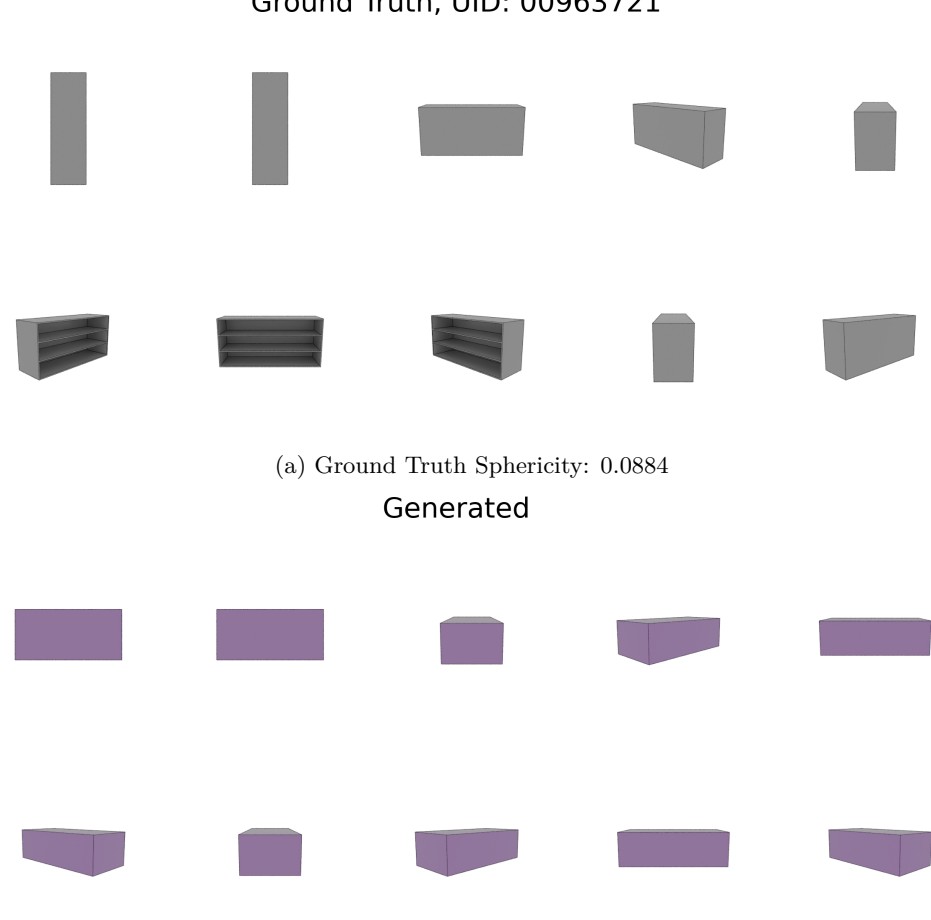

(a) Ground Truth Sphericity: 0.0884

Generated

(b) Generated Sphericity: 0.7255

Figure 20: **SD**: 0.6371; The discrepancy in the sphericity indicates the difference in the overall shape. Higher sphericity indicates closer resemblance to a sphere.

**Discrete Mean Curvature Difference (DMCD)**

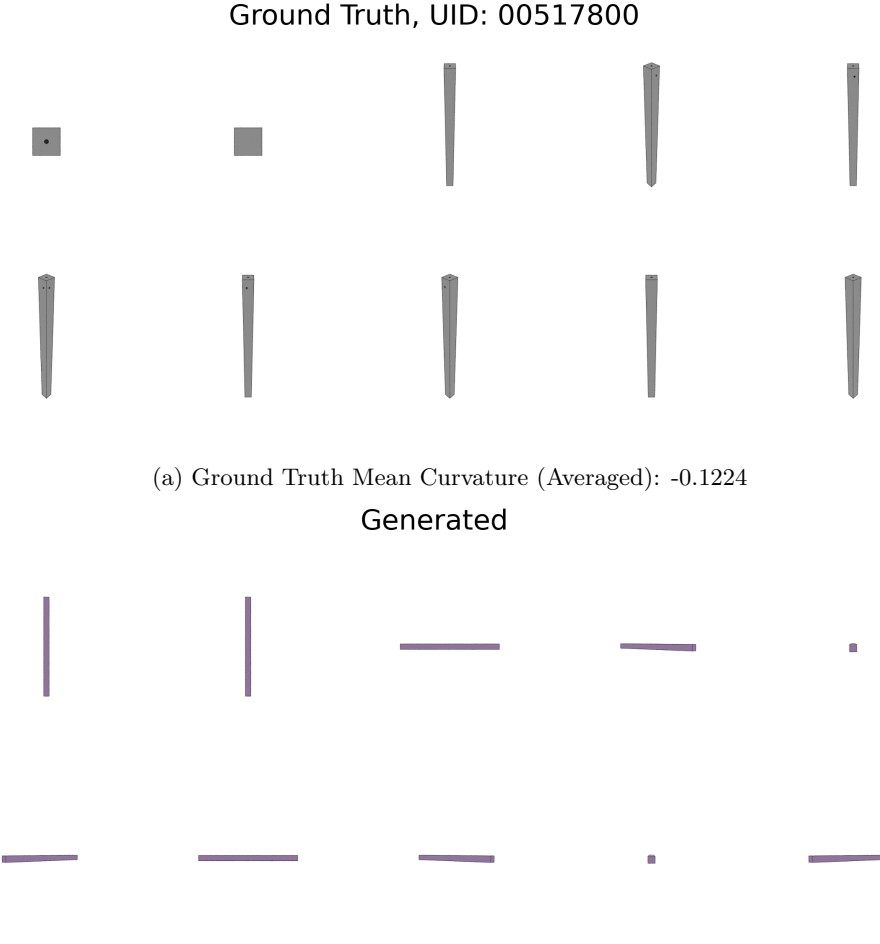

## Ground Truth, UID: 00517800

(a) Ground Truth Mean Curvature (Averaged): -0.1224

## Generated

(b) Predicted Mean Curvature (Averaged): 0.0471

Figure 21: **DMCD**: 0.1061; Discrepancy in *average* mean curvature between the ground truth and generated meshes.

# E    Design Choices

**Annotation Pipeline**    Prior works like Text2CAD utilized ensembles of separate vision and language models. We consciously chose a single, state-of-the-art multimodal model (GPT-4.1) to unify the reasoning process. We hypothesized that a single model with strong instruction-following capabilities would reduce error propagation between the visual perception and text generation stages. We prioritized minimal JSON inputs over verbose serialized formats to maximize the token context window available for reasoning. Consistent with the methodology proposed in Text2CAD, we utilize 10 multi-view images as input for the annotation model.

**Annotation Evaluation**    For evaluating annotation quality, we employed Gemma-3 12B and Mistral Small 3.2 24B. We avoided using GPT-4 for evaluation to prevent self-preference bias, as our annotations were generated by GPT-4.1. By using distinct model families for evaluation, we aimed to decouple the generator's bias from the evaluator's preference.

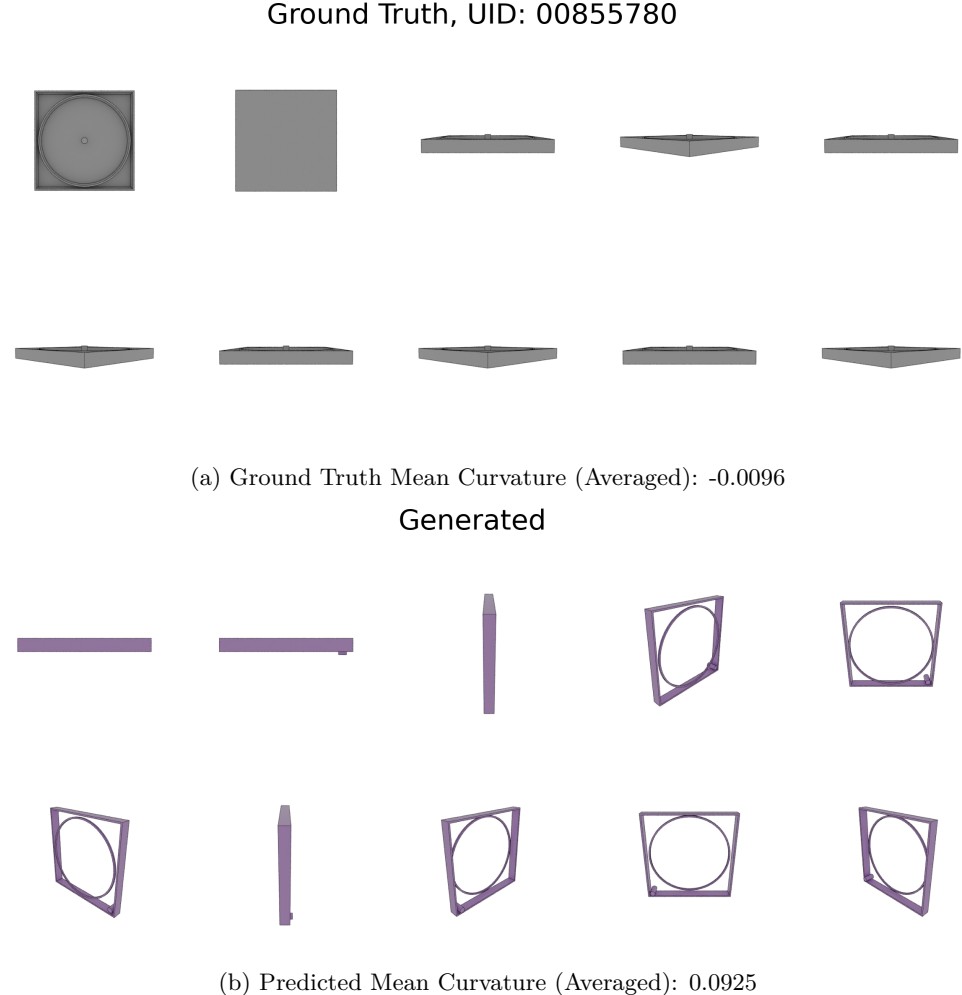

(a) Ground Truth Mean Curvature (Averaged): -0.0096

(b) Predicted Mean Curvature (Averaged): 0.0925

Figure 22: **DMCD**: 0.1021; Discrepancy in *average* mean curvature between the ground truth and generated meshes.

**Fine-tuning**  We selected Qwen-2.5 Coder due to its demonstrated superiority in code generation benchmarks compared to general-purpose LLMs of similar size. For training, we utilized Low-Rank Adaptation (LoRA) with a rank of 64. This rank was chosen empirically; preliminary experiments suggested that lower ranks ($r = 8, 16$) were insufficient for the model to learn the strict syntax of the CAD JSON schema, while full fine-tuning was computationally prohibitive without providing significant performance gains.

**Metrics**  Standard metrics like Chamfer Distance (CD) often fail to capture topological defects in mechanical parts (e.g., a closed hole vs. an open hole). To address this, we utilize Exact Euler Characteristic Match (EECM) and Sphericity Discrepancy (SD). EECM directly reflects the number of holes and connected components, while SD measures the object's deviation from a perfect sphere. Additionally, we employ the Discrete Mean Curvature Difference (DMCD) to calculate the average "mean curvature" across vertices. DMCD is useful for scenarios where surface curvature fidelity is crucial, such as comparing the smoothness of mechanical fillets. Collectively, these metrics are particularly helpful when scaling towards complex 3D object generation, where intricate differences in geometry and topology are critical.

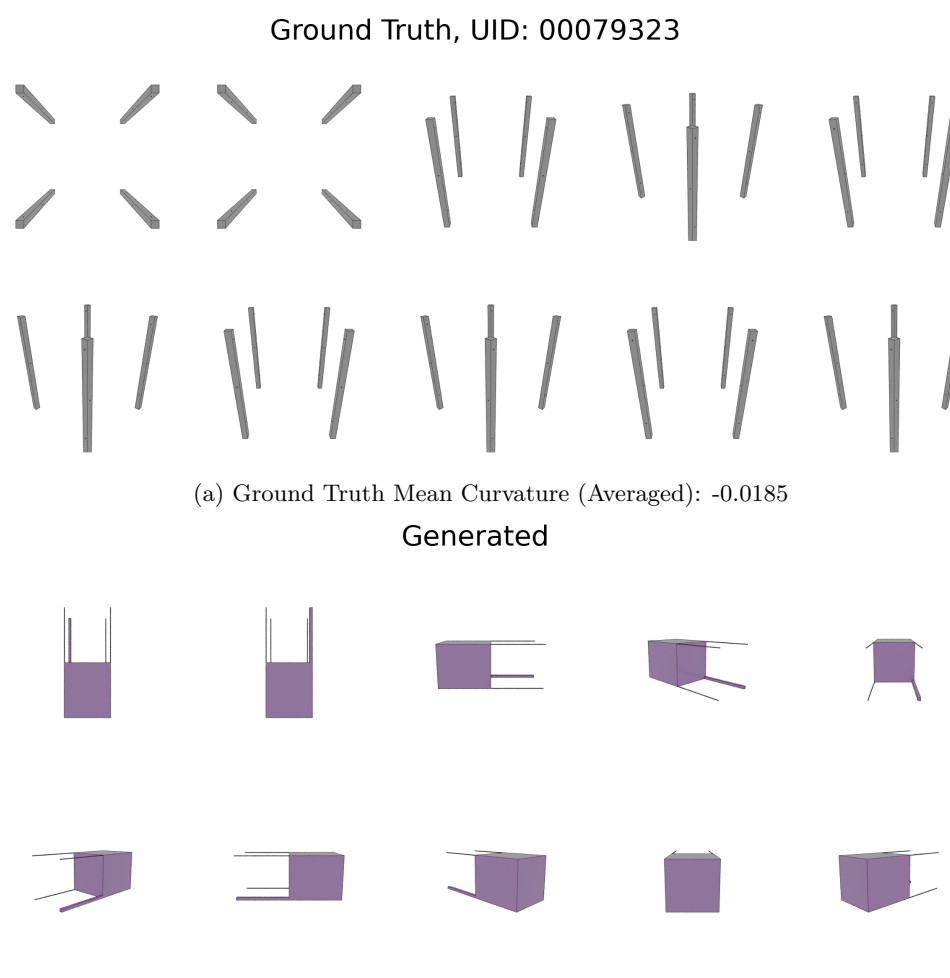

Ground Truth, UID: 00079323

(a) Ground Truth Mean Curvature (Averaged): -0.0185

Generated

(b) Predicted Mean Curvature (Averaged): 0.0778

Figure 23: **DMCD**: 0.0963; Discrepancy in *average* mean curvature between the ground truth and generated meshes.

## F   Licenses for External Assets

This appendix clarifies the licenses and terms of use for the primary external datasets, pre-trained models, and software libraries utilized in this research, based on publicly available information as of May 2025.

The DeepCAD dataset (Wu et al., 2021) is available under the MIT License, as indicated in its official repository. The ABC dataset (Koch et al., 2019), which is a source for DeepCAD, is also distributed under the MIT License. The Fusion 360 Reconstruction dataset (Willis et al., 2021), originating from the Autodesk Fusion 360 Gallery, is referenced in academic literature as being available for research purposes under the Creative Commons Attribution-NonCommercial 4.0 International (CC BY-NC 4.0) license.

The language models leveraged include: Qwen2.5-Coder (Yang et al., 2024), open-source versions of which are provided under the Apache License 2.0. The GPT-4.1 model (OpenAI, 2025) is a proprietary offering from OpenAI, and its usage is governed by OpenAI's prevailing Terms of Use, Usage Policies, and Sharing & Publication Policy. Gemma models (Team et al., 2025), developed by Google, are released under a custom Gemma license.

Key software utilized includes Blender (Blender Online Community, 2025), which is distributed under the GNU General Public License (GPL) version 3 or later. The Python library trimesh (Dawson-Haggerty et al.), used for geometric computations, is available under the MIT License.

The authors have made a best effort to ascertain and respect these licenses and terms. For precise and up-to-date licensing information, direct reference to the official documentation for each asset is recommended.

## G   Licenses for New Assets

Furthermore, the CADmium dataset, which includes the novel annotations generated as part of this research, is released by the authors under the MIT License. The fine-tuned models presented in this paper, which are derived from Qwen2.5-Coder, are released by the authors under the Apache License 2.0.

