# OpenReview forum: "CADmium: Fine-Tuning Code Language Models for Text- Driven Sequential CAD Design"
_TMLR — Accepted by TMLR_

### Review · Reviewer_UbsL · 2025-10-18

**Summary Of Contributions:**

The authors propose a new approach, called CADmium, to generate 3D objects in a text-to-text manner, in which a series of precise object descriptions given in JSON format are prompted to GPT-4.1 to produce corresponding natural language descriptions, and these generated annotations are then used to fine-tune the Qwen2.5-Coder model. The authors also conducted several experiments, including a main comparison with Text2CAD and multiple ablation studies to evaluate the effectiveness of the approach.

**Audience:**

Yes

**Audience Explanation:**

This paper is suitable for the TMLR community as it relates to several active research areas, including LLMs, code generation (albeit only JSON), and ML for Design.

**Claims And Evidence:**

Yes

**Claims Explanation:**

I think the partial claims made by authors are supported by the evidence. Some concerns are given below:

1. **Main claim**: The authors claim that their approach automates sequential CAD design, leverages fine-tuning of code-oriented LLMs, and achieves competitive or improved performance compared to Text2CAD on several metrics. We find that the authors have provided sufficient experimental evidence to support this main claim.
2. **Baseline coverage**: Although the authors compare their model primarily against Text2CAD, I think this work is not merely an exploratory analysis of LLMs ability on a specific problem but rather introduces a new approach that aims to improve both the methodological diversity and performance of LLMs on text-to-CAD generation tasks. Therefore, it would be important to include or at least discuss comparisons with other relevant approaches, such as Text-to-CAD (R. Wang et al., 2025) and CAD-Coder (C. He et al.), to more convincingly demonstrate the advantages of the proposed method.
3. **LLM-as-judge evaluation**: The authors employ an LLM-as-a-judge setup to assess the quality of the newly generated annotations along four perspectives. However, using only Gemma-3-12B as the judge raises concerns about reliability. It would strengthen the study to either employ a more capable LLM as the evaluator or conduct a small-scale human evaluation to calibrate and validate the judgment quality of Gemma-3-12B before using it for large-scale assessment. In current case, relying solely on Gemma-3-12B makes the evaluation less convincing.
4. **Methodological novelty**: From a methodological standpoint, the essence of this approach itself is relatively common in the community. Although this paper may be the first to apply such fine-tuning techniques to CAD generation, the conceptual novelty remains limited. Just a concern although this is not required.

**Requested Changes:**

Please refer to the above comments.

---

> ### Author Response · Authors · 2025-11-16
> **Author response to Reviewer UbsL**
>
> We sincerely thank the reviewer for their detailed feedback and suggestions. We are pleased that the strengths, significance, and clarity of our work have been acknowledged. We are also grateful that the extensive nature of our experiments and the relevance of the newly proposed metrics are appreciated. We have carefully considered the reviewer’s comments and concerns and addressed them below. All changes made to the manuscript are highlighted in blue text in the attached PDF submission.
> ___
> ***Baseline coverage***
>
> We agree with the reviewer’s feedback and acknowledge that recent works have explored a variety of strategies for data generation, CAD sequence representation, and LLM- and MLLM-based fine-tuning approaches for automating CAD. Published concurrently with or shortly after CADmium's submission, other works focused on avenues such as CAD editing, hierarchical CAD sequence representation, and learning from feedback. CADFusion (*R. Wang et al. 2025*) and CAD-Coder (*A.C. Doris et al. 2025*) are a few such examples. We have highlighted the fundamental differences between our approach and those below.
> 1. CAD-Coder is primarily designed as an Image-to-CAD solution, utilizing a VLM fine-tuned to generate executable CadQuery Python code (which is different from JSON-based representation) directly from images. CADmium, on the other hand, focuses exclusively on text-to-CAD generation where the input is a natural language description.
> 2. CADFusion explores a different training strategy that combines sequential learning with visual feedback learning with DPO, while our pipeline relies on SFT.
>
> CADmium’s methodology and experiments also uniquely stand out in terms of the simplicity of directly using JSON-based commands as text inputs, cross-dataset evaluations and model size ablations. Further, while recent CAD generation. As future work, we wish to integrate some of the advancements in recent works with CADmium’s training methodology in terms of multimodality and controllability. We have edited the related work and discussion section to include additional information about this.
> ___
> ***LLM-as-a-Judge Evaluation***
>
> We agree with the reviewer’s comment and agree that multi-judge or human evaluation can further strengthen reliability. A human evaluation study was not feasible within current resources and given time constraints. However, to address this concern, we included an additional independent LLM-as-a-judge evaluation using Mistral-Small-3.2-24B-Instruct, a newer model from a distinct model family. This reduces single-model bias and helps validate the consistency of the Gemma-3-12B results. Figure 2 is updated accordingly. Further, Text2CAD annotations were originally produced with a Mistral model, so this complementary evaluation reduces the risk that our results favor GPT-style outputs.
> ___
> ***Methodological novelty***
>
> We agree with the reviewer that our main contribution is not a new model architecture. Our work’s primary contribution is the reframing of text-to-CAD generation as a text-to-code task. Instead of designing another specialized and complex architecture from scratch, we show that this domain can be successfully tackled by adapting powerful, general-purpose code LLMs with no modifications. We believe this paradigm shift is a significant contribution, as it offers a simpler, more scalable, and more pragmatic approach that directly benefits from the rapid progress in foundation models.

---

### Review · Reviewer_7RcQ · 2025-10-25

**Summary Of Contributions:**

**Contributions:**

1. **Annotation pipeline and proposed dataset**: The authors proposed a pipeline that uses GPT-4.1's multimodal capabilities to generate textual descriptions for training. The work re-annotated the DeepCAD dataset with high-quality, human-like descriptions for 176,017 CAD models. They presented this new large-scale dataset with improved annotations compared to existing Text2CAD descriptions.

2. **Reformulation of the CAD task**: This work propose to formulate the task as a purely text-to-text problem, minimizing manual intervention. For this reformulated task, the authors fine-tuned a Qwen-2.5-Coder using LoRA on the composed dataset to generate JSON-formatted CAD sequences from natural language descriptions.

3. **Proposing new evaluation metrics**: The paper introduces several new geometric and topological metrics including Sphericity Discrepancy (SD), Discrete Mean Curvature Difference (DMCD), Exact Euler Characteristic Match (EECM), and watertightness assessment to provide richer structural insights beyond traditional point-cloud metrics.

**Strengths:**
1. The text-to-text formulation for CAD generation is novel and well positioned for using existing pre-trained models effectively.
2. The proposed dataset is beneficial to the community. The work also provides comprehensive evaluation of the proposed model on multiple datasets including the proposed one. I believe the dataset and the evaluation metrics can be valuable tools and resources to the research community working on 3D object generation and CAD automation.
3. CADmium shows better performance in reconstructing specific geometry from text comparing to baselines. It achieves much higher F1 scores than Text2CAD for arcs and circles. The authors also use the Sphericity Discrepancy metric to show that CADmium can better capture the object's overall compactness.

**Weaknesses:**
- The performance degradation when evaluating ob different annotation styles shows room for improvement in the pipeline design. The model struggles on out-of-distribution data, which suggests that it could be overfitting to the specific style of its training prompts.

**Audience:**

Yes

**Audience Explanation:**

1. **Computer Vision and 3D Processing Community**: The work addresses text-conditioned 3D object generation, which is an active area of research. The novel evaluation metrics (sphericity, curvature, Euler characteristic) provide valuable tools for the 3D vision community.

2. **Applied AI and Engineering**: This work makes it easier to automate CAD design using AI, which is important for industry. People who work on engineering or manufacturing with AI will find it useful. It speeds up the creation process and lets people who are not CAD experts use the system without knowing special CAD programming.

**Broader Impact Concerns:**

- Quality and safety are serious consideration. Ensuring the reliability and correctness of generated designs is crucial. Errors in the generated CAD files can have significant consequences if deployed in real-world applications. In that sense, the labor requested for validation is non-negligible and should be considered.

**Claims And Evidence:**

Yes

**Claims Explanation:**

1. The paper provides detailed evaluation across multiple datasets including Text2CAD, CADPrompt, Fusion360 and comparisons to other baseline models. The work also provides ablation studies on model scale and annotation quality.
2. The introduced geometric and topological metrics e.g. SD, DMCD, EECM, watertightness are mathematically sound. The metrics are clearly defined with clear mathematical formulations and implementation details.

**Requested Changes:**

1. The authors should more clearly position the contributions as significant incremental improvements upon prior work. E.g. the re-annotation of the DeepCAD and Fusion360 datasets using an adapted pipeline. The performance gains should be largely attributable to the pre-trained Qwen-2.5 Coder base model, which is not a novel contribution of this work. This should be clearly distinguished from the paper's contribution.

2. Manual validation is encouraged as the "LLM-as-a-judge" evaluation has been questioned a lot in the field, which often considered methodologically flawed. Using Gemma-3 12B to judge GPT-4.1's output could be preference leakage, as the judge is likely biased towards the style of the LLMs. The claims of superior "human-likeness" and "clarity" are less reliable without independent human validation.

3. A more comprehensive ablation on annotation sytle is encouraged, comparing annotations from more MLLMs not just Text2CAD. In section 4.3 This result suggests that the models may be fitting on the specific style and vocabulary of their training data, rather than learning a truly generalized, style-agnostic "understanding" of geometric instructions.

---

> ### Author Response · Authors · 2025-11-16
> **Author response to Reviewer 7RcQ**
>
> We sincerely thank the reviewer for their detailed feedback and suggestions. We are pleased that the strengths, significance, and clarity of our work have been acknowledged. We are also grateful that the extensive nature of our experiments and the relevance of the newly proposed metrics are appreciated. We have carefully considered the reviewer’s comments and concerns and addressed them below. All changes made to the manuscript are highlighted in blue text in the attached PDF submission.
> ___
> ***The authors should more clearly position the contributions as significant incremental improvements upon prior work.***
>
> We agree with the reviewer’s feedback and have revised the introduction and conclusion to better position our original and incremental contributions. We addressed the performance attribution in two key ways:
>
> 1. We added a clear statement to the introduction (Page 2) that the competitive performance from SFT is largely attributed to the inherent, pre-trained qualities of the Qwen 2.5-Coder base model.
> 2. We highlight in the introduction that our GPT-4.1-based pipeline represents an improvement over the Text2CAD annotation pipeline. Our advanced pipeline leverages multimodal capacity to generate better annotations for the highly utilized DeepCAD and Fusion360 datasets, which is detailed further in Figure 1, and Section 3.1.
> ___
> ***Manual validation is encouraged as the "LLM-as-a-judge" evaluation has been questioned a lot in the field***
>
> We agree that human evaluation is the gold standard, but running a sufficiently large human study for 10,000 paired annotations is beyond our current resources. To address the concern about potential preference leakage when using Gemma-3 12B as the judge, we added an additional independent LLM-as-a-judge evaluation using Mistral-Small-3.2-24B, a newer model from a distinct model family. Figure 2 and Section 3.1 are updated accordingly. Importantly, Text2CAD annotations were originally produced with a Mistral model, so this complementary evaluation reduces the risk that our results favor GPT-style outputs.
> ___
> ***A more comprehensive ablation on annotation style is encouraged, comparing annotations from more MLLMs not just Text2CAD***
>
> The scope of this study is limited to expert-only annotations (i.e., detailed descriptions and steps to construct the CAD object). This reflects our goal of assessing the instruction-following capacity essential for design experts in the CAD field, rather than targeting general users who input simplified or abstract descriptions lacking precise measurements and geometric details. Such expert-level data is crucial for unambiguously reconstructing objects, which aligns with engineering requirements. At the time of writing, only Text2CAD had released a public dataset consisting of expert-level annotations.
>
> We did an analysis of other proposed text annotation datasets following Text2CAD.
>
> 1. CADLLM/TCADGen (*J. Liao et al. 2025*) proposed a semi-automated annotation pipeline with human and LLM-based quality inspections in the pipeline. They pass multi-view images, point clouds, and the CAD parameters independently through different LLMs/VLMs and obtain micro and macro-level descriptions. We have updated Figure 2 in the draft to include this dataset in the evaluation.
> 2. The dataset proposed by CAD-MLLM (*OmniCAD, J. Xu et al. 2025*) used InternVL2-26B to generate simpler prompts (e.g. “Generate a CAD model with a minimalist, modern design featuring a tall, vertical panel supported by a simple, geometric base. The base consists of a rectangular platform with a protruding block on one side, providing stability and a unique visual element. The vertical panel is sleek and unadorned, emphasizing clean lines and a contemporary aesthetic.”). Similarly, the dataset from CADFusion (*R. Wang et al. 2025*) contains only beginner-level prompts.
> 3. CAD-LLaMA (*J. Li et al. 2025*) proposed a new hierarchical data annotation pipeline that uses GPT-4o to generate component-level and macro-level detailed descriptions for each object. However, this dataset is not publicly accessible at present.
> ___
> Lastly, regarding the weakness of generalization pointed out by the reviewer, the paper acknowledges and highlights the necessity for further investigation into the model's generalization capabilities across diverse prompt types in Section 5.

---

### Review · Reviewer_W47D · 2025-11-02

**Summary Of Contributions:**

The paper introduces CADmium, a text-to-CAD system that reframes CAD generation as a text-to-text translation problem. To support this, the authors build a large dataset of approximately 176K CAD models with expert-style textual descriptions synthesized using GPT-4.1. They fine-tune Qwen2.5-Coder models to convert natural-language prompts into JSON-based CAD command sequences and show consistent improvements over Text2CAD, particularly in geometric accuracy and shape integrity. CADmium introduces new geometry-focused evaluation measures, including Sphericity Discrepancy, Euler characteristic matching, and Discrete Mean Curvature Difference, complementing standard fidelity metrics. Experiments across DeepCAD, CADPrompt, and Fusion360 benchmarks demonstrate superior geometric element fidelity and better performance on curvature and topological metrics, establishing CADmium as an effective text-to-CAD generation framework.

**Audience:**

Yes

**Audience Explanation:**

The topic is highly relevant to the TMLR community, particularly researchers working on program synthesis, text-to-3D generation, CAD automation, and leveraging LLMs for structured geometry tasks. The paper makes contributions in dataset creation, fine-tuning code-oriented LLMs for CAD generation, and introducing new geometric/topological evaluation metrics — areas that intersect machine learning, computer-aided design, and generative modeling.

**Broader Impact Concerns:**

This work focuses on automating CAD design from natural language, which has significant positive potential for accessibility, rapid prototyping, engineering productivity, and democratizing 3D design.

**Claims And Evidence:**

Yes

**Claims Explanation:**

The paper presents extensive quantitative evidence across several datasets (DeepCAD, CADPrompt, Fusion360) and evaluates on both model-generated and human-written instructions. The reported metrics include geometric and topological measures, invalidity rates, F1 scores, Chamfer distances, and newly introduced structural metrics. Confidence intervals and statistical significance tests strengthen the credibility of comparisons. Ablation on model scale (1.5B → 14B) and cross-training experiments using both CADmium and Text2CAD architectures provide convincing evidence that the approach is effective and that improvements arise from both data quality and model design.

**Requested Changes:**

While the paper presents a compelling dataset and text-to-CAD pipeline, the manuscript would benefit from a deeper discussion and clearer positioning of the data generation methodology and data selection strategy for CAD code. In particular:

1- Clarify the data generation technique: Expand the explanation of how GPT-4.1’s multimodal reasoning was leveraged beyond basic captioning — e.g., how the pipeline balances geometric fidelity with natural language fluency and avoids over-constrained descriptions that simply mirror JSON keys. A flow diagram or pseudo-prompt example could strengthen transparency.

---

> ### Author Response · Authors · 2025-11-16
> **Author response to Reviewer W47D**
>
> We sincerely thank the reviewer for their detailed feedback and suggestions. We are pleased that the strengths, significance, and clarity of our work have been acknowledged. We are also grateful that the extensive nature of our experiments and the relevance of the newly proposed metrics are appreciated. We have carefully considered the reviewer’s concerns and addressed them below. All changes made to the manuscript are highlighted in blue text in the attached PDF submission.
> ___
> ***Clarify the data generation technique***
>
> We appreciate the reviewer’s suggestion. We would like to clarify that the paper already includes
> 1.  The complete GPT-4.1 prompts in Appendix Figures 9–14, showing exactly how the minimal JSON, multi-view renders, and instructions are combined, and
> 2. A workflow diagram in the main text (Figure 1) that illustrates the full data-generation pipeline.
>
> These materials provide more details than pseudo-prompts or summaries, enabling full reproducibility.

---

### Decision · Action_Editor_RmhQ · 2025-12-09

**Recommendation:** Accept with minor revision

**Additional Comments:**

The reviewers found the proposed CADmium approach novel, the evaluation extensive and comprehensive, and the dataset and evaluation metrics compelling and valuable to the community. On the other hand, some concerns have been raised regarding the position of contributions and novelty, the "LLM-as-a-judge" evaluation, the inclusion of more baselines and comprehensive ablation, and clarification of the data generation process. The authors have provided concise responses to address these concerns and revised the manuscript accordingly.

After the rebuttal, this work received mixed ratings (1 Leaning Accept, and 2 Leaning Reject). **Reviewer UbsL** votes to weakly accept this work after reading the author's response. In the official recommendation, **Reviewer 7RcQ** remains unconvinced that the revisions sufficiently strengthen the draft, mainly due to: 1) The lack of human validation is still a significant methodological gap which cannot be resolved by adding a second LLM judge due to the intrinsic bias and preference leackage; 2) The ablation on annotation style remains limited, especially on the concern of overfitting to specific prompt styles rather than learning generalizable geometric understanding; and 3) Many design choices throughout the pipeline appear intuitive (the annotation generation process, metric selection, training hyperparameters, and LLM-as-a-judge choice), and a deeper investigation would be beneficial. In a confidential comment to AE, **Reviewer W47D** believes that while this work presents a well-executed system and demonstrates solid engineering effort, the approach largely relies on existing high-capacity proprietary models and standard LoRA fine-tuning, without introducing fundamentally new algorithms, training paradigms, or theoretical insights. As such, the work feels more like a data-centric pipeline contribution rather than a significant advancement in ML methodology.

I read this paper in detail and agree with all reviewers that the proposed CADmium approach and the high-quality CAD annotation dataset are both valuable contributions to the community, while the raised and remaining concerns on human/ LLM-as-a-judge validation, ablation on annotation style, deeper investigation on the design choices, and incremental improvements upon existing work are also valid and reasonable.

The acceptance criteria of TMLR are based on the positive answers to the following two questions: 1) Are the claims made in the submission supported by accurate, convincing, and clear evidence? and 2) Would some individuals in TMLR's audience be interested in the findings of this paper?. Given that **all reviewers consistently vote "yes" to both of these two questions** in their review comments as well as in their official recommendation (which I totally agree with), I recommend **accepting this work, but with a "minor" (somewhat like a major) revision**.

To further address the concerns raised by the reviewers and improve the quality of this work, I have the following suggestions.

**(Sub)section of Limitation:** Add a (sub)section of limitation to explicitly and clearly discuss the limitations of the proposed method in detail, such as the lack of human validation and the potential drawbacks of the current LLM-as-judge validation, the current scope on expert-only annotations and the challenges for generalization to other annotations (e.g., simplified or abstract descriptions from general users).

**More Discussion on the Design Choices:** Add more and deeper discussion, with analysis if appropriate, on the design choices throughout the CADmium pipeline, such as the annotation generation process, metric selection, training hyperparameters, and LLM-as-a-judge choice.

**Better Positioning of the Contributions:** If possible, find a way to further clearly and explicitly distinguish the contributions/novelty of the proposed approach from previous work. I believe the proposed CADmium pipeline itself with the high-quality dataset is already a good contribution to TMLR.

In the camera-ready version, I expect the authors to properly address the concerns raised above. In addition, please ensure that all discussions are carefully incorporated into the paper and release the fine-tuned model and the CADmium dataset as promised.

**Audience:**

Yes

**Audience Explanation:**

All reviewers believe some individuals in TMLR's audience could be interested in the findings of this paper.

**Claims And Evidence:**

Yes

**Claims Explanation:**

This work proposes CADmium, an LLM-based approach for text-based sequential CAD generation. The key contributions are two-fold: 1) a novel annotation pipeline powered by GPT-4.1 to generate high quality CAD annotations from JSON descriptions and multi-view images, which leads to a large-scale dataset of 176K CAD annotations; 2) a fine-tuned Qwen 2.5-Coder-14B model, which is trained on the proposed dataset with SFT, to generate JSON-formatted CAD sequences solely from natural language prompts. Experimental results demonstrate that the proposed CADmium approach can successfully automate the CAD design process with pure natural language descriptions.

All reviewers believe the claims made in this paper are supported by convincing and clear evidence. Please refer to the comment below for more details.